# Bringing Light to the Threshold: Identification of Multi-Score Regression Discontinuity Effects with Application to LED Manufacturing

## Abstract

The regression discontinuity design (RDD) is a widely used framework for threshold-based causal effect estimation in causal inference. Recent extensions incorporating machine learning (ML) adjustments have made RDD an appealing approach for researchers utilizing causal ML toolkits. However, many real-world applications, such as production systems, involve multiple decision criteria and logically connected thresholds, necessitating more sophisticated identification strategies, which are not clearly addressed in the recent literature. We derive a novel identification result for the complier effect in the multi-score RDD (MRD) setting by extending unit behavior types to multiple dimensions. Further, we show that under mild assumptions, this identification result does not depend on subsets of units with constant response. We apply our findings to simulated and real-world data from opto-electronic semiconductor manufacturing, employing estimators that adjust for covariates through machine learning. Our results offer insights into enhancing current production policies by optimizing the cutoff points, demonstrating the applicability of MRD in a manufacturing context.

## 1 Introduction

The Regression Discontinuity Design (RDD) is a quasi-experimental strategy for the identification and estimation of causal effects that has been widely applied in empirical economics (Hartmann et al., 2011; Card et al., 2015; Flammer, 2015; Calvo et al., 2019), public policy (Lee, 2008) and the social sciences (Angrist & Lavy, 1999). Its appeal lies in its ability to deliver credible causal estimates under minimal assumptions, utilizing discontinuities in treatment assignment rules that are typically based on an observed running or score variable (Imbens & Lemieux, 2008; Lee & Lemieux, 2010). Typical examples are credit scores, GPAs or vote shares. The classic RDD setting assumes that treatment assignment hinges on a single, continuous forcing variable crossing a known threshold. When correctly specified, this setting enables identification of average treatment effects for units near the cutoff (Cattaneo et al., 2019). In particular, RDD can be applied even if typical causal machine learning assumptions such as *unconfoundedness* (Rubin, 1974) or *positivity* (Austin, 2011) do not hold (Imbens & Lemieux, 2008).

However, in many real-world contexts – particularly in industrial, operational, or engineering systems – treatment decisions are based not on a single score, but on multiple criteria (Sabaei et al., 2015). Such situations call for an extension of the classical RDD setup to multi-score RDD (MRD), in which treatment is assigned when a combination of variables jointly satisfies a threshold condition (Papay et al., 2011). Recent work has surveyed MRD (Reardon & Robinson, 2012; Wong et al., 2013; Porter et al., 2017) and proposed estimators for test score-based (An et al., 2024) and geographical (Keele & Titiunik, 2015) applications.

This paper addresses the identification and estimation challenges that arise in the context of multi-score RDD, particularly in complex decision-making environments such as production systems. In such systems, decision rules are often implemented through a layered combination of logic and thresholds applied to multiple inputs, making the treatment assignment mechanism more intricate

than in the standard RDD case. We develop new identification results tailored for multi-score RDD settings. Specifically, we provide a formal framework that defines and categorizes behavioral types of units. We investigate general boolean-type cutoff rules, such as "AND"-type and "OR"-type rules, commonly observed in MRD environments and analyze how their properties influence local identification. Further, we show how the complier effect can be identified using these categories. We demonstrate the utility of our framework by applying it to a real-world problem in light-emitting diode (LED) production, where treatment assignment is based on multiple quality indicators. Using real-world manufacturing data, we estimate the causal effect of a threshold-based production policy and show how the estimated effects can guide the optimal tuning of the decision threshold to improve manufacturing outcomes. We complement our empirical results with a simulated study using a semi-synthetic clone of the production environment, highlighting the value of multi-score RDD for counterfactual analysis and industrial policy design.

**Our contributions** to the causal machine learning literature are twofold. On the theoretical side, we advance the multi-dimensional regression discontinuity (MRD) framework, previously developed in the econometric literature. Specifically, we provide a rigorous definition and categorization of unit behavior types in the multi-cutoff case, introducing in particular the novel class of *indecisive units*, which has no counterpart in the one-dimensional setting. We further establish an identification theorem for the complier effect, derive precise conditions under which non-changing units (alwaystakers and nevertakers) can be rejected, and show how our identification results enable recovery of the complier effect in the multi-dimensional case. On the practical side, we illustrate the applicability and usefulness of the MRD framework in a real-data industrial application, thereby demonstrating the potential of causal machine learning methods in practice.

## 2 BACKGROUND

### 2.1 RELATED LITERATURE

Early theoretical work on extending RDD to **Multi-Score Regression Discontinuity Designs** recognized that such designs require new identification strategies (Papay et al., 2011). While this setting included multiple treatment levels, most recent papers on MRD study the identification of a binary treatment that is assigned based on multiple indicators, using either "AND" connections of rules (Choi & Lee, 2018; Keele & Titiunik, 2015), "OR" connections (Wong et al., 2013), or both (Reardon & Robinson, 2012; Imbens & Zajonc, 2009). An important special case involves geographical MRD, which features a two-dimensional design with longitude and latitude as scores (Keele & Titiunik, 2015; Cattaneo et al., 2025). See Appendix D for an overview of the MRD estimators typically studied in recent literature (Porter et al., 2017). Additionally, recent work has introduced new estimators based on a minimax approach (Imbens & Wager, 2019) and on decision trees (Liu & Qi, 2024). Porter et al. (2017) note that although there is a rich body of studies using MRD, there is a lack of theoretical understanding of MRD estimators.

Our work further contributes to a growing field of **applications of causal machine learning in management and operations**. Calvo et al. (2019) use a one-dimensional fuzzy RD design in public infrastructure projects. Hünermund et al. (2021) highlight the value of causal machine learning for business decision-making and provide an overview of methods, including RDD. Mithas et al. (2022) give an overview of RDD applications in operations management. Schacht et al. (2023) study policy making in semiconductor manufacturing using propensity score-based estimation in the double machine learning framework. Vuković & Thalmann (2022) investigate the development of research on causal discovery in manufacturing, focusing on motivation, common application scenarios, impact, and implementation challenges. Finally, there have been applications of causal machine learning to policy learning in different business fields, e.g., supply chain management (Wyrembek et al., 2025) or marketing (Huber, 2024). Despite these recent advances, the application of causal machine learning methodologies within operational and industrial domains remains underexplored.

### 2.2 RDD SETTING

RDD dates back to a study by Thistlethwaite & Campbell (1960) on scholarship programs. Recent sources on RDD often rely on identification and estimation results by Hahn et al. (2001), as well as prominent surveys (Imbens & Lemieux, 2008; Lee & Lemieux, 2010; Cattaneo et al., 2019). RDD

consists of three key ingredients: A score $X$ that rates the individuals; a cutoff $c$ that splits the support of the score into two groups; and a treatment $D$, which is assigned to one of the groups based on their score and the cutoff, $D_i = \mathbb{1}[X_i \geq c]$ (Cattaneo et al., 2019).

The parameter of interest is the average treatment effect at the cutoff $c$, $\mathbb{E}[Y(1) - Y(0) \mid X = c]$ with $Y(1)$ and $Y(0)$ being the potential outcomes of the individuals (Rubin, 2005). The central RDD assumption is continuity.

**Assumption 1.** *Continuity. The conditional mean of the potential outcomes $\mathbb{E}[Y_i(d) \mid X_i = x]$ for $d \in \{0, 1\}$ is continuous at the cutoff level $c$.*

Under Assumption 1, RDD provides inference around the threshold as plausible as that from a randomized experiment (Lee, 2008). The average treatment effect at the threshold $\tau_0 = \mathbb{E}[Y_i(1) - Y_i(0) \mid X_i = c]$ is identified as $\tau_0 = \lim_{x \to c^+} \mathbb{E}[Y_i \mid X_i = x] - \lim_{x \to c^-} \mathbb{E}[Y_i \mid X_i = x]$ (Hahn et al., 2001). The basic RDD estimator runs separate local linear regressions on each side of the cutoff:

$$\hat{\tau}_{\text{base}}(h) = \sum_i w_i(h) Y_i,$$

where the $w_i(h)$ are local linear regression weights that depend on the data through the realizations of the running variable only, and $h > 0$ is a bandwidth.

Under standard conditions (Hahn et al., 2001), which include that the running variable is continuously distributed, and that the bandwidth $h$ tends to zero at an appropriate rate, the estimator $\hat{\tau}_{\text{base}}(h)$ is approximately normally distributed in large samples, with bias of order $h^2$ and variance of order $(nh)^{-1}$, with sample size $n$.

More recent work, which we will also employ in the empirical section, uses a flexible covariate adjustment based on potentially nonlinear adjustment functions $\eta$. The estimator takes the following form:

$$\widehat{\tau}_{\text{RDFlex}}(h; \eta) = \sum_i w_i(h) M_i(\eta), \quad M_i(\eta) = Y_i - \eta(Z_i). \tag{1}$$

Here, $\eta$ is the influence of $Z$ on the outcome $Y$ and is estimated using ML methods (Noack et al., 2024).

## 3 General Identification Strategies in Multi-score RDD

In this section, we give a formal definition of common behavior types of units (e.g., complier, defier) in multi-score, two-stage decision settings that employ cutoff-rules for the initial treatment assignment. We derive an identification result using these unit categories. In particular, we show that under certain assumptions, the identification does not depend on subsets of unit types with constant response. This is a new result in the literature and our main theoretical contribution.

In practical settings, the easiest way to improve a complex cutoff-rule is to analyze and adjust cutoffs individually, e.g., one can estimate the effect on complier with respect to a specific subrule involving a single cutoff. Taking the AND-rule $D := \mathbb{1}[X_1 > c_1] \mathbb{1}[X_2 > c_2]$ as an example, one can analyze the effect on complier of $G := \mathbb{1}[X_1 > c_1]$ at the cutoff to gain insights on how to improve $c_1$. This can be achieved in different ways: First, using a fuzzy setting in which we regard $G$ as the assigned and $D$ as the actual treatment. Second, by conditioning on the complying units and using a sharp estimator. Thus, knowledge of unit behavior can open up a different way of estimating the complier effect.

Expanding on the previous example, now suppose that the treatment assignment $T := \mathbb{1}[X_1 > c_1] \mathbb{1}[X_2 > c_2]$ is known, while the final decision rule $D$ is unknown and does not always comply with $T$. Thus, the question arises how knowledge of units that comply with $T$ in the final treatment $D$ can improve the estimation of the effect of $G := \mathbb{1}[X_1 > c_1]$ on the treatment outcome.

### 3.1 Setup

For each individual $i$, let $X_i = (X_{1,i}, \ldots, X_{K,i}) \in \mathbb{R}^K$ denote the score variables. Further, given $c \in \mathbb{R}^K$, let $I_i(c) := (I_{1,i}(c_1), \ldots, I_{K,i}(c_K))$ denote the corresponding indicator variables with

$I_{k,i}(c_k) := \mathbb{1}[X_{k,i} > c_k]$ and let the observed outcome for individual $i$ be $Y_i$. We regard the entries $I_i(c)$ as boolean variables and allow any composition of AND, OR and negation operations $(\wedge, \vee, \overline{\square})$ over this set of atoms to form general boolean functions $g(I_i(c))$.

**Definition 1.** *A mapping $T : \mathbb{R}^K \to \{0,1\}$ is called a decision rule. We say that a decision rule $T$ is a cutoff rule (on $\mathbb{R}^K$) if there exists a boolean mapping $g$ such that $T_i = T(X_i) = g(I_i)$.*

With slight abuse of notation, we use $T(c)$, $T(X_i \mid c)$ and $T_i(c)$ to indicate the use of a specific cutoff $c \in \mathbb{R}^K$. Note that

$$T(X + \epsilon \mid c) = T(X \mid c - \epsilon) \quad \text{and} \quad T(\lambda X \mid c) = T\left(X \,\Big|\, \frac{c}{\lambda}\right) \tag{2}$$

holds for $\epsilon \in \mathbb{R}^K, \lambda \in \mathbb{R}_{>0}$. Thus, without loss of generality, we assume that the cutoff of interest is $c = 0$. We suppose that $Y_i$ depends in the following way on a cutoff rule $T$ and a general decision rule $D$:

$$Y_i = Y_i(T_i, D_i) = \left(Y_i(0,0)(1 - D_i) + Y_i(0,1)D_i\right)(1 - T_i)$$
$$+ \left(Y_i(1,0)(1 - D_i) + Y_i(1,1)D_i\right)T_i,$$

where $T$ is the treatment assignment and $D$ is the actually implemented treatment. Unless otherwise stated, in the proofs we make no further assumptions on $D$ except that it is a decision-rule.

## 3.2 UNIT CATEGORIZATION

Given this setup, there are certain groups of individuals that are especially interesting, namely the nevertaker, the alwaystaker, the complier and the defier with respect to the pair $(T, D)$ or $(G, D)$, where $G$ is a subrule of $T$. We follow the intuition that changes in the cutoff, or equivalently in the observed score values – relevant for $T$ – are necessary to categorize the behavior of a unit $i$.

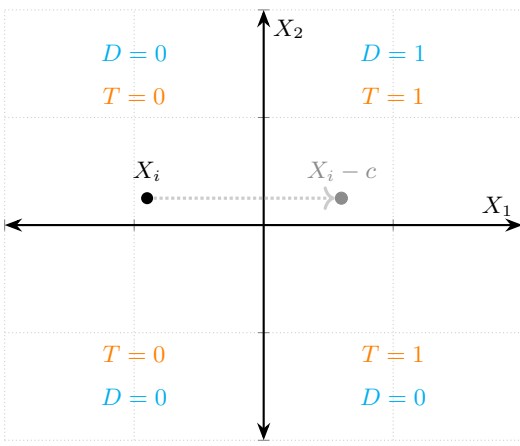

Figure 1: Cutoff rules $D = I_1 \wedge I_2$ and $T = I_1$. The decision boundaries coincide with the coordinate axes. When $X_2 > 0$, $T$ complies with $D$; otherwise, $D = 0$ regardless of the value of $X_1$, which governs the behavior of $T$.

Figure 1 illustrates this idea for $D := I_1 \wedge I_2$ and $T := I_1$, and shows that only changes $c \in \mathbb{R}^2$ that affect $T$ are relevant for categorizing the behavior of a unit $i$ with respect to $(T, D)$. Additional variables that affect only $D$ may not be controllable or even observable. The following definition formalizes the intuition behind relevant directions of change in a cutoff rule.

**Definition 2.** *Let $T$ be a cutoff rule on $\mathbb{R}^K$. Denote with $e_1, \ldots, e_K$ the unit direction in $\mathbb{R}^K$ and with*

$$S(T) := \left\{k \,\big|\, \exists c \in \mathbb{R}^K, \lambda \in \mathbb{R} : T(0 \mid c + \lambda e_k) \neq T(0 \mid c)\right\}$$

*the support directions of $T$. The support of $T$ is defined as the linear hull over the support directions:*

$$\text{supp}(T) := \left\{ \sum_{k \in S(T)} \lambda_k e_k \,\middle|\, \lambda_k \in \mathbb{R}, \, k \in S(T) \right\}$$

The above notion of the support of a cutoff rule is motivated by the existence of a change along a single coordinate direction. It is almost trivial to see that an empty support also excludes changes along multiple directions, justifying the definition.

**Lemma 1.** *$T$ is constant if and only if $S(T) = \emptyset$, or equivalently, if and only if $\text{supp}(T) = \{0\}$.*

From now on, we require that $T$ is not degenerate in the sense that $\text{supp}(T) \neq \{0\}$.

For the introductory AND-rule example, $D := I_1(c) \wedge I_2(c)$ and $T := I_1(c)$ we have $\text{supp}(T) = \mathbb{R} \times \{0\}$ and $\text{supp}(D) = \mathbb{R}^2$. If a unit $i$ complies in the assigned treatment $T$ with the actual $D$, one would require that $T_i$ produces the same output as $D_i$, even under any[1] hypothetical change of the cutoff $c_1$. In other words, both rules should be synchronous on the support of $T$, which is the case if $X_{2,i} > c_2$. Thus, it is $X_{2,i}$ (or equivalently $c_2$) that controls the behavior of $i$. To make this distinction more apparent, it is useful to have a notation for entries $X \in \mathbb{R}^K$ that do not affect $T$:

$$N^T := \left\{ X \in \mathbb{R}^K \,\middle|\, T(X \,|\, c) = T(0 \,|\, c) \text{ for all } c \in \mathbb{R}^K \right\}$$

In particular, one can show the following.

**Proposition 1.** *For each $X \in \mathbb{R}^K$, there exists a unique decomposition $X = X^T + X^{\perp T}$ with $X^T \in \text{supp}(T)$ and $X^{\perp T} \in N^T$. The orthogonal projection $P_T(X) := \sum_{k \in S(T)} \langle X, e_k \rangle e_k$ onto $\text{supp}(T)$ satisfies the above properties, where $\langle \cdot, \cdot \rangle$ denotes the standard scalar product on $\mathbb{R}^K$.*

That is, according to Proposition 1, one has the following decomposition

$$\mathcal{S} := \mathbb{R}^{|S(T)|} \times \mathbb{R}^{K - |S(T)|} \simeq{}^2 \text{supp}(T) \oplus N^T = \mathbb{R}^K$$

of the score space into two distinct parts. The former part captures the decision changes of $T$ and the latter consists of free variables not affecting $T$, but potentially controlling the unit category. This observation allows for a general definition of unit categories.

**Definition 3.** *Let $i$ be a unit and $X_i = X_i^T + X_i^{\perp T}$. Then $i$ is said to be a nevertaker (an alwaystaker) of $T$ with respect to $D$ iff $D\left(X_i^{\perp T} - c\right) = 0$ (iff $D\left(X_i^{\perp T} - c\right) = 1$) for all $c \in \text{supp}(T)$.*
*Further, $i$ is said to be a complier (a defier) of $T$ with respect to $D$ iff $T\left(0 \,|\, c\right) = D\left(X_i^{\perp T} - c\right)$ (iff $T\left(0 \,|\, c\right) \neq D\left(X_i^{\perp T} - c\right)$) for all $c \in \text{supp}(T)$.*
*Let the sets of compliers, nevertakers, alwaystakers, and defiers be denoted by $\text{ComP}(T, D)$, $\text{Nt}(T, D)$, $\text{At}(T, D)$, and $\text{DeF}(T, D)$, respectively.*

Particularly, if $D$ is a cutoff-rule one obtains the following more intuitive equivalencies, which capture the notion of a simultaneous cutoff changes along relevant directions.

**Proposition 2.** *Let $D$ be a cutoff rule over $\mathbb{R}^K$ and $i$ be a unit. Then $i$ is a nevertaker (an alwaystaker) of $T$ w.r.t. $D$ iff $D_i(c) = 0$ (iff $D_i(c) = 1$) for all $c \in \text{supp}(T)$. Further, $i$ is a complier (a defier) of $T$ w.r.t. $D$ iff $T_i(c) = D_i(c)$ (iff $T_i(c) \neq D_i(c)$) for all $c \in \text{supp}(T)$.*

Note that the terminology introduced in Definition 3 indeed introduces well-defined categories:

**Proposition 3.** *The sets $\text{At}(T, D)$, $\text{Nt}(T, D)$, $\text{ComP}(T, D)$ and $\text{DeF}(T, D)$ are pairwise disjoint.*

From now on, whenever we assume that $D$ is a cutoff rule, we restrict ourselves to the case in which $D$ has at least as much information for decision-making as $T$. This means that $D$ might depend on $I_{k,i}(0)$ for $k \in S(T)$. In addition, we suppose that $D$ does not depend on $I_{k,i}(c)$ with $c \neq 0$ for $k \in \text{supp}(T)$, effectively restricting to the case where both decision rules have a zero cutoff. [3]

---

[1]A more refined, local definition could relax this requirement to "any reasonable changes". For ease of presentation, we stick to the global version.

[2]Both linear spaces are isomorphic; $\oplus$ denotes the direct sum.

[3]This restricts the general case. Suppose that the decision maker $D$ is an opportunist and ignores all the other scores as long as a certain incentive $X_k$ exceeds an even higher cutoff $0 < c_k < X_k$. Only then would $D$ comply with $T$.

Even for cutoff rules $D$ under these restrictions, the categories above are not exhaustive if $\dim(\mathrm{supp}(T)) > 1$. For example, let $D := (\overline{I}_1 \wedge I_2)$ and $T := I_1 \wedge I_2$. Then $\mathrm{supp}(T) = \mathbb{R}^2$ and for $T(0\,|\,0) = D(0\,|\,0)$ but $0 = T(0\,|\,(-1,1)) \neq D(0\,|\,(-1,1)) = 1$. Thus, $D$ is not constant nor equal to $T$ or $\overline{T}$ on the support of $T$. We call individuals of this remaining category *indecisive* and denote *the set of indecisives (of $T$ with respect to $D$)* by $\mathrm{Ind}(T, D)$.

Figure 2 visualizes the general case in which $D$ is not a cutoff rule.

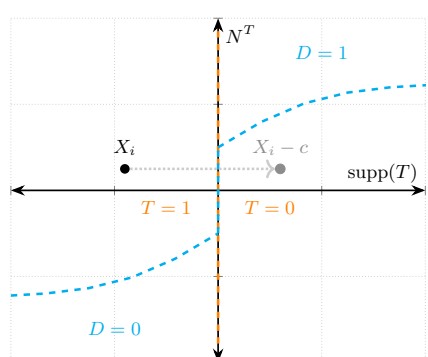 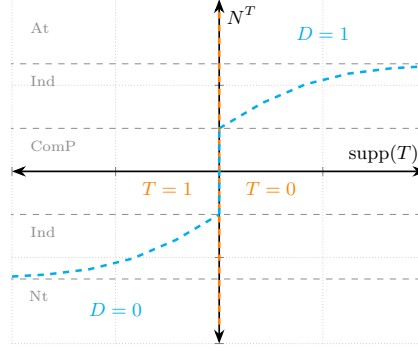

(a) No matter which change $c$ of $X_i$ in the $\mathrm{supp}(T)$ plane we imagine, the responses of $D$ and $T$ are equal. A shift of score $X_i$ by $c$ is equivalent to a simultaneous shift of the decision boundaries by $c$, which amounts to a coordinate shift for $D$ and a cutoff shift for $T$ motivating Definition 3.

(b) Unit categorizations with respect to $(T, D)$. While $\mathrm{supp}(T)$ captures the directions in which $T$ can change, unit behavior is determined by $N^T$. The space is partitioned (from top to bottom) into areas of alwaystakers, indecisives, compliers, indecisives and nevertakers of $T$ with respect to $D$.

Figure 2: General case with $D$ not being a cutoff rule. The decision boundaries of $T$ and $D$ are indicated with orange and blue dashed lines.

Note that given $T$ with $\dim(\mathrm{supp}(T)) > 1$, one can always construct a cutoff rule $D$ that exhibits indecisive items. At least one has the following:

**Proposition 4.** *Let $D$ be a cutoff rule on $\mathbb{R}^K$ and let $i$ denote an individual. If $\dim(\mathrm{supp}(T)) = 1$ then $i \in \mathrm{At}(T, D) \cup \mathrm{Nt}(T, D) \cup \mathrm{DeF}(T, D) \cup \mathrm{ComP}(T, D)$.*

Moreover, our definitions of complier, nevertaker and alwaystaker imply the corresponding definitions in (Imbens & Lemieux, 2008). For this, let $c^+, c^- \in \mathrm{supp}(T)$ be two directions that induce change in $T$, that is:

$$\lim_{\lambda \to 0} T(X_i \,|\, \lambda c^+) = 1 \text{ and } \lim_{\lambda \to 0} T(X_i \,|\, \lambda c^-) = 0$$

Using the above complier definition:

$$T(X_i \,|\, c) = T(X_i^T \,|\, c) = T(0 \,|\, c - X_i^T) = D(X_i^{\perp T} + X_i^T - c) = D(X_i - c)$$

for $c \in \mathrm{supp}(T)$. In particular, $T$ and $D$ coincide in a neighborhood of zero, and thus

$$\lim_{\lambda \to 0} D_i(\lambda c^+) = \lim_{\lambda \to 0} D(X_i - \lambda c^+) = 1 \text{ and } \lim_{\lambda \to 0} D_i(\lambda c^-) = \lim_{\lambda \to 0} D(X_i - \lambda c^-) = 0$$

where $D_i(c) := D(X_i - c)$ for $c \in \mathrm{supp}(T)$. The corresponding statements for nevertaker and alwaystaker follow analogously. Indeed, requiring consistency in the limit for any direction one would arrive at a local definition of the unit categories sufficient for the multi-score RDD setting. Before continuing with the identification part we are going to expand on the AND-rule example.

**Example (AND-Rules)** Let $D := \bigwedge_{j=1}^{K} I_j$ and $T := \bigwedge_{j=1}^{k} I_j$ for $k \in \{1, \ldots, K-1\}$. Then $\mathrm{supp}(T) = \mathbb{R}^k \times \{0\}^{K-k}$. Further, we have the following unit categorizations: **(i)** $\mathrm{ComP}(T, D) = \{i \,|\, \forall k < j \leq K : X_{j,i} > 0\}$ Note that $X_{j,i} > 0$ for all $k < j \leq K$ is equivalent to $\wedge_{j=k+1}^{K} I_{j,i}(c)$ being one for $c \in \mathrm{supp}(T)$. Thus, the rule $D_i$ effectively reduces to $T_i$. **(ii)** $\mathrm{At}(T, D) = \emptyset$ Since for each $X_i \in \mathbb{R}^K$ there exists a $c \in \mathrm{supp}(T)$ such that $D_i(c) = 0$. For example choose any $c$ with $X_{1,i} \leq c_1$ and $c_j = 0$ for $j \neq 1$. **(iii)** $\mathrm{Nt}(T, D) = \{i \,|\, \exists k < j \leq K : X_{j,i} \leq 0\}$ Note that $i$ being in the set on the right is equivalent to $\wedge_{j=k+1}^{K} I_{j,i}(c)$ being zero for all $c \in \mathrm{supp}(T)$. This is equivalent to $D_i(c) = 0$ for all $c \in \mathrm{supp}(T)$, since given an $i$ one can always find a cutoff $c \in \mathrm{supp}(T)$

such that $T_i(c) = 1$, which implies that one of the indicators $I_{j,i}$ for $j > k$ has to be zero. **(iv)** $\mathrm{DeF}(T, D) = \emptyset$ Note that $T_i(c) = 0$ implies $D_i(c) = 0$, and that $c \in \mathrm{supp}(T)$ can always be chosen such that the former is satisfied.

Additional examples can be found in Appendix A.2 illustrating instances of the remaining unit categories (excluding the indecisive case). We conclude that the introduced definitions are indeed reasonable.

### 3.3 Effect Identification

Inspired by the work of (Hahn et al., 2001; Imbens & Angrist, 1994), we use the introduced unit categories to prove an identification theorem for the effect on complier at the cutoff. For this section, we require that the outcome $Y$ does not directly depend on the treatment assignment $T$. Denote the set of all unit categories by

$$\mathcal{C} := \{\mathrm{ComP}(T, D),\, \mathrm{Nt}(T, D),\, \mathrm{At}(T, D),\, \mathrm{DeF}(T, D),\, \mathrm{Ind}(T, D)\}$$

and the set of non-change categories by $\mathcal{C}^0 := \{\mathrm{Nt}(T, D),\, \mathrm{At}(T, D)\}$. We assume that the categorization of a unit is independent of the support part of $T$ in a neighborhood of the cutoff, that is:

**Assumption 2.** *There exists an $\epsilon > 0$ such that $\Pr\left(i \in \mathrm{Cat} \,\middle|\, X_i^T = x\right) = \Pr\left(i \in \mathrm{Cat} \,\middle|\, X_i^T = 0\right)$ for $\|x\| \leq \epsilon$ and $\mathrm{Cat} \in \mathcal{C}$.*

Note that this requirement relates to the independence assumptions used in (Imbens & Angrist, 1994). We further rely on the following local continuity assumption, which is a variation of the standard continuity assumption (Assumption 1):

**Assumption 3.** *There exists an $\epsilon > 0$ such that $x \mapsto \mathbb{E}(Y_i(d) \,|\, X_i^T = x,\, i \in \mathrm{Cat})$ is continuous for $\|x\| \leq \epsilon$, $d \in \{0, 1\}$ and $\mathrm{Cat} \in \mathcal{C}$.* [4]

Two more assumptions are required to make further use of the continuity. First, we deny the existence of indecisive items, since this category does not allow structured conclusions about $D$ based on knowledge of $T$.

**Assumption 4.** $\mathrm{Ind}(T, D) = \emptyset$

Second, we assume that the directions $x^+$, $x^- \in \mathrm{supp}(T)$ along which we aim to estimate the complier effect induce a change in $T$.

**Assumption 5.** $1 = \lim_{\lambda \to 0} T\left(\lambda x^+ \,|\, 0\right) \neq \lim_{\lambda \to 0} T\left(\lambda x^- \,|\, 0\right) = 0$

This assumption is implicit in one-dimensional RDD designs and imposes no real restriction in practice, as $T$ is generally assumed to be known. The proof of the following proposition can be found in Appendix A.3.

**Theorem 1.** *Let Assumptions 2, 3, 4 and 5 hold. Then the complier effect at the cutoff is identified as*

$$\mathbb{E}\left(Y_i(1) \,\middle|\, X_i^T = 0,\, i \in \mathrm{ComP}\right) - \mathbb{E}\left(Y_i(0) \,\middle|\, X_i^T = 0,\, i \in \mathrm{ComP}\right) =$$

$$\frac{1}{\Pr\left(i \in \mathrm{ComP} \,\middle|\, X_i^T = 0\right)} \left(\lim_{\lambda \to 0} \mathbb{E}\left(Y_i \,\middle|\, X_i^T = \lambda x^+\right) - \lim_{\lambda \to 0} \mathbb{E}\left(Y_i \,\middle|\, X_i^T = \lambda x^-\right)\right) - C$$

*with $C$ being the correction term for defier:*

$$C := \frac{\Pr\left(i \in \mathrm{DeF} \,\middle|\, X_i^T = 0\right)}{\Pr\left(i \in \mathrm{ComP} \,\middle|\, X_i^T = 0\right)} \left(\mathbb{E}\left(Y_i(0) \,\middle|\, X_i^T = 0,\, i \in \mathrm{DeF}\right) - \mathbb{E}\left(Y_i(1) \,\middle|\, X_i^T = 0,\, i \in \mathrm{DeF}\right)\right)$$

This identification result allows for two immediate conclusions. First, one is free to choose among the directions $x^+$ and $x^-$ satisfying Assumption 5. Second, the proof suggests that dropping subsets $\Omega \subset \mathrm{Nt} \cup \mathrm{At}$ does not affect identification, as long as doing so does not violate Assumptions 3 and 2. We provide a corresponding result in Appendix A.3 (see Theorem 2). We call estimates of the complier effect of $T$ when removing $\Omega$ the *sub-set complier effect of $G$ excluding $\Omega$*.

---

[4]This assumption can be weakened by assuming only directional continuity, which would render the effect dependent on the chosen directions.

## 4 APPLICATION SETTING

In the following section, we will bridge the gap between our theoretical considerations and the empirical setting studied. We consider an inline rework process in lot-based opto-electric semiconductor manufacturing during the phosphor conversion step. Production lots, each consisting of 784 individual LEDs, that fail to achieve a certain quality level are subjected to an additional rework step to improve overall yield. For details on the conversion process, we refer to related work (Cho et al., 2017; Schwarz et al., 2024).

The score consists of two components: The *distance score*, $X_D$, measures the distance from the mean color point $C := (C_x, C_y)$ of a lot to the optimal target in the color space. The *yield improvement score*, $X_Y$, is a relative measure that evaluates the quality distribution of individual chips in a lot by calculating a hypothetical scenario in which the target is ideally met by the mean of the lot. If there is high variability in the quality of parts within a lot, this improvement is negligible or even negative. Figure 3 depicts the score components in more detail.

The treatment is assigned by a binary "AND" decision rule $T = I_D \wedge I_Y$. We assume that the final decision makers (i.e., human operators) $D$ have an informational advantage regarding the improvement score and that they use this advantage to override $T$, while remaining cautious about possible degradation. This cautious operator assumption aligns with the observed one-sided fuzziness in the $I_Y$ dimension and the strict compliance to the distance rule $I_D$, as observed in real data (see Figure 4). In particular, whenever $T$ suggests that a rework step should be carried out, the operator

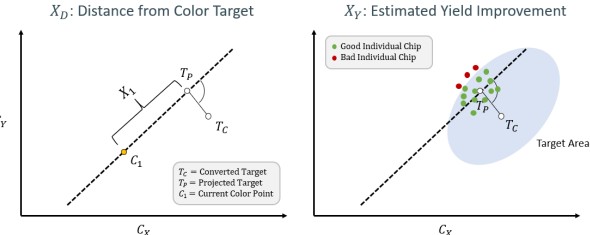
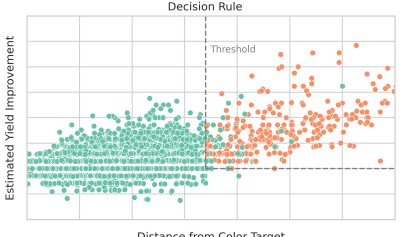

Figure 3: Left: $X_D$ is defined as the distance between the current mean color point and the target point. Right: $X_Y$ evaluates the expected improvement by calculating the share of in-specification chips in the lot. This is done by moving the current distribution of color points to the target.

Figure 4: Real data plot w.r.t. the score components $X_D$ and $X_Y$. The decision boundary $T$ is dashed. The actual treatment assignment (green and orange) follows an unobserved rule $D$, rendering the MRD fuzzy.

may override it due to the informational advantage not captured in $T$. Conversely, if $T$ does not suggest a rework treatment, we assume that the operator accepts this decision, being cautious of possible degradation. We formalize the inclination toward this negative override into an additional score variable $X_{op}$ that captures the information advantage: $D = T \wedge I_{op}$. A DGP based on the cautious operator assumption is outlined in Appendix E. In this case, we have:

$$\mathrm{ComP}(T, D) = \{i \,|\, X_{op,i} > 0\} \text{ and } \mathrm{Nt}(T, D) = \{i \,|\, X_{op,i} \leq 0\},$$

as well as

$$\mathrm{ComP}(G, D) = \{i \,|\, X_{op,i} > 0, X_Y > 0\} \text{ and } \mathrm{Nt}(G, D) = \{i \,|\, X_{op,i} \leq 0\} \cup \{i \,|\, X_{Y,i} \leq 0\}$$

for the sub-rule $G := I_D$ of $T$ (see Example 3.2). Employing Theorem 2 from Appendix A one can estimate the sub-set complier effect of $G$ excluding the nevertaker $\Omega := \{i \,|\, X_{Y,i} \leq 0\}$ instead of the complier effect of $G$. [5]

## 5 EFFECT ESTIMATION WITH RDD

We benchmark estimators in accordance to Section 3 on both semi-synthetic and real-world data as described in Section 4. We compare estimators without adjustment, with conventional adjustment, penalized linear adjustment and adjustment using a stacked ensemble learner.

---

[5] Since the condition $X_{Y,i} \leq 0$ is global, the required continuity and stability assumptions are likely to be satisfied in practical applications.

## 5.1 SEMI-SYNTHETIC DATA

To provide a realistic benchmark of the estimators across different settings, we generate semi-synthetic data following Algorithm 1[6] The process is calibrated to match real data characteristics. We draw $n = 10,000$ observations and repeat the experiment $r = 250$ times. The oracle value is estimated using a local linear kernel regression on the differences in true potential outcomes. The covariates consist of statistics describing the quality of individual items.

We evaluate the cut-offs $c_D$ and $c_Y$ separately, estimating the complier effect of $G \in \{I_Y, I_D\}$ as identified in Theorem 1, the **subset** complier effect of $G$ given its counterpart (in accordance to Theorem 2 and intent-to-treat estimates with and without the subset conditioning. The complier effects are estimated under a fuzzy design; the intent-to-treat effects under a sharp design.

As shown in Figure 5, the intent-to-treat oracles are closer to zero than the complier effects due to the inclusion of individuals who are nevertakers with respect to each cutoff rule. As visible in 5a, for $I_D$ we estimate an overall negative effect, although it is not significant at $95\%$-level. The subset effect for the fuzzy case exhibits a smaller bias as the percentage of nevertakers in the estimation sample is smaller. The estimated effect remains unchanged as only nevertakers but no compliers were removed. This underlines our theoretical argument. For the intent-to-treat estimator, a higher share of compliers in the subsample increases the estimated effect of treatment rule $G$. The subset estimators have a comparable estimated variance (see Table 4 in Appendix C.1), with the coverage overall appearing slightly more credible. Generally, the covariate adjustment reduces the standard error in the estimation, especially for the sharp estimators.

The estimates for $I_Y$ are small and positive. The fuzzy estimator on the full data has a high standard error, which increases further with ML adjustment. This may be due to a small jump in treatment probability in the full data, destabilizing the ML estimate. The subset estimator along this axis removes much more observations within the bandwidth of the estimation thus decreasing the variance. Additional results can be found in Appendix C.

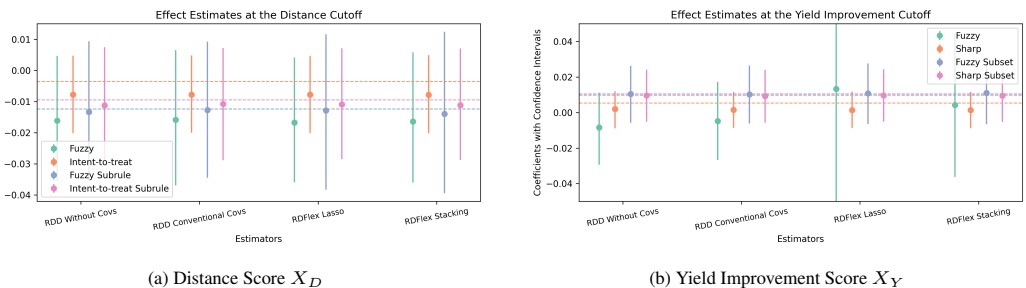

(a) Distance Score $X_D$        (b) Yield Improvement Score $X_Y$

Figure 5: Median coefficient and median CI for fuzzy and intent-to-treat estimators along $X_D$ (left) and $X_Y$ (right) with simulated data. The different colors depict estimators on the full sample and on the subset. The dashed line shows the oracle estimate for each setting, with the complier (fuzzy) oracles overlapping.

## 5.2 REAL DATASET

This section presents results from estimation on the real data with $n = 9,103$ observations from the production system. Additional to information on color measurements, we include the shopfloor workload. Our primary interest in the real system is evaluating $I_D$.

As shown in Table 1 on the real data, we estimate a positive effect. The subset estimates have a lower variance, as we remove uncertainty due to nevertakers according to $X_Y$. In case of the intent-to-treat estimators, the addition of ML adjustment improves the variance of the estimation. For the fuzzy estimators, there is no clear improvement.

The comparison in Figure 6 provides insight into the differing effect signs between semi-synthetic and real data. While the semi-synthetic process (left) appears to have a rework threshold closer than the optimal distance, the real data (right) suggests a threshold that is further away, resulting in a significant outcome jump at the current threshold.

---

[6]The python implementation of this process will be made available.

| setting | method | Coef | s.e. | CI 2.5% | CI 97.5% | RMSE left | Log loss left | RMSE right | Log loss right | % s.e. change |
|---|---|---|---|---|---|---|---|---|---|---|
| Fuzzy | RDD Conventional Covs | 0.1031 | 0.0508 | 0.0035 | 0.2028 | | | | | 22.2532 |
| | RDD Without Covs | 0.0387 | 0.0416 | -0.0428 | 0.1202 | | | | | 0.0000 |
| | RDFlex Lasso | 0.0999 | 0.0470 | -0.0004 | 0.2003 | 0.1452 | 0.0141 | 0.1523 | 0.6142 | 12.9249 |
| | RDFlex Stacking | 0.1097 | 0.0500 | 0.0023 | 0.2171 | 0.1429 | 0.0188 | 0.1570 | 0.6322 | 20.1333 |
| Fuzzy on Subset | RDD Conventional Covs | 0.0630 | 0.0426 | -0.0205 | 0.1466 | | | | | 36.5324 |
| | RDD Without Covs | 0.0369 | 0.0312 | -0.0243 | 0.0981 | | | | | 0.0000 |
| | RDFlex Lasso | 0.0713 | 0.0304 | 0.0031 | 0.1395 | 0.1410 | 0.0209 | 0.1507 | 0.5424 | -2.5667 |
| | RDFlex Stacking | 0.0768 | 0.0319 | 0.0048 | 0.1487 | 0.1394 | 0.0121 | 0.1530 | 0.5018 | 2.1538 |
| Sharp | RDD Conventional Covs | 0.0393 | 0.0208 | -0.0015 | 0.0802 | | | | | 2.2467 |
| | RDD Without Covs | 0.0165 | 0.0204 | -0.0235 | 0.0564 | | | | | 0.0000 |
| | RDFlex Lasso | 0.0357 | 0.0167 | -0.0026 | 0.0740 | 0.1425 | | 0.1513 | | -17.9717 |
| | RDFlex Stacking | 0.0367 | 0.0168 | -0.0018 | 0.0753 | 0.1408 | | 0.1530 | | -17.5203 |
| Sharp on Subset | RDD Conventional Covs | 0.0362 | 0.0205 | -0.0040 | 0.0764 | | | | | -2.3944 |
| | RDD Without Covs | 0.0098 | 0.0210 | -0.0314 | 0.0509 | | | | | 0.0000 |
| | RDFlex Lasso | 0.0364 | 0.0184 | -0.0031 | 0.0758 | 0.1408 | | 0.1525 | | -12.2509 |
| | RDFlex Stacking | 0.0307 | 0.0181 | -0.0093 | 0.0708 | 0.1395 | | 0.1540 | | -14.0691 |

Table 1: Coefficients, standard errors, and quality of fit for the different estimators for the distance dimension in the real dataset.

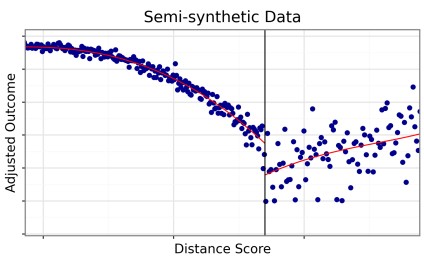

(a) RDD plot for the semi-synthetic process.

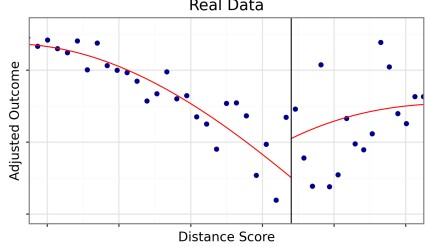

(b) RDD plot for the real production data.

Figure 6: Comparison of local linear regressions around the cutoff for the semi-synthetic and real data. The opposite sign of the estimated effect can be explained by the calibration of the semi-synthetic process. Specifically, the modeling of the operator decision could be imperfect by taking only incomplete information into account. Particularly, the real-world operator might have more knowledge, such as machine states and outputs, as well as practical job experience.

## 6 CONCLUSION AND LIMITATIONS

Our paper presented a novel application of RDD in the context of threshold-based decision-making in industrial manufacturing. By integrating multi-score RDD techniques with recent advancements in causal machine learning – particularly flexible covariate adjustment – we demonstrated how to evaluate threshold based decision policies on real data.

Our formalization of unit behavior categories (complier, nevertaker, alwaystaker, defier and indecisive units) in multi-dimensional cutoff rules yields a novel effect identification result in MRD settings. Among the required assumptions, the local stability of the unit categories (Assumption 2) is the most debatable one. As it involves counterfactual reasoning about potential category changes, it cannot be verified using observed data. This shortcoming is shared with related independence assumptions common in RDD literature. Causal identification is not possible without any of these assumptions. The question is whether such an assumption is interpretable enough to justify an approximate conformance in empirical studies. Compared to previously presented formulations, we draw from the intuitive language of unit categories to aid the argument for or against applicability.

## 7 REPRODUCIBILITY STATEMENT

We provide full source code and detailed instructions in the supplementary material to reproduce all numerical experiments. The data-generating processes for synthetic experiments are fully specified. Information about runtime environments and computing resources is documented in the Appendix. Hyperparameters used in the experiments are listed in the relevant sections and in the supplementary material. The implementation of our semi-synthetic process is included and will also be released as part of an open-source causal inference package to facilitate community use. The dataset used in the real-data application is subject to data protection regulations and therefore cannot be released.

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

# APPENDIX A  PROOFS

## A.1  UNIT CATEGORIZATION

**Lemma 1.** *$T$ is constant if and only if $S(T) = \emptyset$ or equivalently if and only if* $\operatorname{supp}(T) = \{0\}$.

Before proofing the above statement let us make some simple observations. Let
$$S(T \,|\, X) := \{k \,|\, \exists \lambda \in \mathbb{R} \,:\, T(X \,|\, 0) \neq T(X \,|\, 0 + \lambda e_k)\}$$
denote the set of local change directions of $X$. Then one has the following properties:

1. $S(T \,|\, X)$ depends only upon the quadrant of $X$, since within a quadrant the $I_k$'s and the $\overline{I}_k$'s are constant.

2. $S(T) = \bigcup_{X \in \mathbb{R}^K} S(T \,|\, X)$
   Since $T(0 \,|\, c + \lambda e_k) \neq T(0 \,|\, c)$ is equivalent to $T(-c \,|\, 0 + \lambda e_k) \neq T(-c \,|\, 0)$.

3. $S(T \,|\, X)$ can be empty.
   For example let $T = (I_1 \wedge I_2) \vee I_3$ and $X_j > 0$ for $j = 1, 2, 3$. Then $S(T \,|\, X) = \emptyset$, since $T$ is true as long as there are at least two indicators that are true. Changing the cutoff in only one direction $k$ does not affect $T(0 + \lambda e_k)$. This example can be extended such that changes in multiple directions do not affect $T$.

*Proof.* One direction is clearly trivial. For the other suppose $T$ is not constant. Then one has $0 = T(X \,|\, 0) \neq T(\hat{X} \,|\, 0) = 1$ for appropriate $X, \hat{X} \in \mathbb{R}^K$. Define the sequence

$$X^k := X + \sum_{j=1}^{k} \left\langle \hat{X} - X, \, e_j \right\rangle e_j \quad 0 \leq k \leq K$$

for $0 \leq k \leq K$. Then $X^K = \hat{X}$, $X^0 = X$ and $X^k - X^{k-1} = \lambda_k e_k$ with $\lambda_k := \left\langle \hat{X} - X, \, e_k \right\rangle$. We assume that $S(T \,|\, X^k) = \emptyset$ for all $k$. Then

$$T(X^{k-1} \,|\, 0) = T(X^{k-1} \,|\, 0 - \lambda_k e_k) = T(X^k \,|\, 0)$$

and thus per induction $0 = T(X \,|\, 0) = T(\hat{X} \,|\, 0) = 1$. Which contradicts the assumption. Thus, there exists some $k$ with $S(T \,|\, X^k) \neq \emptyset$. Which shows that $S(T) \neq \emptyset$. $\qquad\square$

As a general assumption for the next statements we require that $T$ is not trivial, that is $\operatorname{supp}(T) \neq \{0\}$.

**Proposition 1.** *For each $X \in \mathbb{R}^K$ there exists a unique decomposition $X = X^T + X^{\perp T}$ with $X^T \in \operatorname{supp} T$ and $X^{\perp T} \in N^T$. The orthogonal projection $P_T(X) := \sum_{k \in S(T)} \langle X, \, e_k \rangle e_k$ onto* $\operatorname{supp}(T)$ *satisfies the above properties.*

*Proof.* Using equations 2 one can see that $X + \lambda Y \in N^T$ for $X, Y \in N^T$ and $\lambda \in \mathbb{R}_{>0}$. Further $T(-X \,|\, c) = T(0 \,|\, c + X) = T(X \,|\, c + X) = T(X - X \,|\, c) = T(0 \,|\, c)$ for $X \in N^T$ and $c \in \mathbb{R}^K$. Thus, $N^T$ is a linear subspace of $\mathbb{R}^K$. Let $k \in S(T)$ then there exists $\lambda \in \mathbb{R}$, $c \in \mathbb{R}^K$ such that $T(0 \,|\, c + \lambda e_k) \neq T(0 \,|\, c)$. This means that $-\lambda e_k \notin N$. Since $N^T$ is a linear space this means that $e_k$ can also not be in $N$. Now let $k \notin S(T)$. Then $T(0 \,|\, c) = T(0 \,|\, c - e_k) = T(e_k \,|\, c)$ for all $c \in \mathbb{R}^K$ and thus $e_k \in N^T$. This shows that $\operatorname{supp}(T) \cap N^T = \{0\}$. Let $X^T := \sum_{k \in S(T)} \langle X, \, e_k \rangle e_k$ be the projection on to the subspace $\operatorname{supp}(T)$ and $X^{\perp T} := X - X^T$. Suppose there exists some $c$ such that $T(X^{\perp T} \,|\, c) \neq T(0 \,|\, c)$. Let

$$c^k := c - \hat{X} + \sum_{j=1}^{k} \lambda_j e_j \quad 0 \leq k \leq K$$

with $\lambda_j := \left\langle X^{\perp T}, \, e_j \right\rangle$. Then $c^0 = c - X^{\perp T}$, $c^K = c$ and $c^k = c^{k-1} + \lambda_k e_k$ holds. Further, one has
$$T(0 \,|\, c^k) = T(0 \,|\, c^{k-1} + \lambda_k e_k) = T(0 \,|\, c^{k-1})$$
for all $1 \leq k \leq K$. Otherwise, $k \subset S(T)$ and thus $\lambda_k = 0$ by definition of $X^{\perp T}$. Using induction we derive $T(X^{\perp T} \,|\, c) = T(0 \,|\, c^0) = T(0 \,|\, c^K)$ which contradicts the assumption. For uniqueness suppose there exists another decomposition $X = Z^T + Z^{\perp T}$ with the above properties. Then $Z^T - X^T = Z^{\perp T} - X^{\perp T} \in \operatorname{supp}(T) \cap N^T = \{0\}$. $\qquad\square$

**Proposition 2.** *Let $D$ be a cutoff rule over $X$. A unit $i$ is*

    *1. a nevertaker (of $T$ with respect to $D$) iff $D_i(c) = 0$*

    *2. an alwaystaker (of $T$ with respect to $D$) iff $D_i(c) = 1$*

    *3. a complier (of $T$ with respect to $D$) iff $T_i(c) = D_i(c)$*

    *4. a defier (of $T$ with respect to $D$) iff $D_i(c) \neq T_i(c)$*

*for all $c \in \mathrm{supp}(T)$.*

*Proof.* Note that $T_i(c) = T_i(X_i \,|\, c) = T(X_i^T \,|\, c) = T(0 \,|\, c - X_i^T) = T(0 \,|\, \hat{c})$ and $D_i(c) = D(X_i - c \,|\, 0) = D(\hat{X}_i - \hat{c})$ with $\hat{c} := c - X_i^T \in \mathrm{supp}(T)$ by applying equation (2) and Proposition 1. $\qquad\square$

**Proposition 3.** *The sets $\mathrm{At}(T, D)$, $\mathrm{Nt}(T, D)$, $\mathrm{ComP}(T, D)$ and $\mathrm{DeF}(T, D)$ are pairwise disjoint.*

*Proof.* It is easy to see that $\mathrm{At}(T, D) \cap \mathrm{Nt}(T, D) = \emptyset$ and $\mathrm{ComP}(T, D) \cap \mathrm{DeF}(T, D) = \emptyset$. For $i \in C := \mathrm{Nt}(T, D) \cup \mathrm{At}(T, D)$ it follows that $D(X_i^{\perp T} - c)$ is constant for all $c \in \mathrm{supp}(T)$. Suppose that $i \in C \cap \mathrm{ComP}(T, D)$. The latter would mean, that $T(0 \,|\, c) = D(X_i^{\perp T} - c)$ for all $c \in \mathrm{supp}(T)$. This contradicts $\mathrm{supp}(T) \neq \{0\}$. Now suppose that $i \in C \cap \mathrm{DeF}(T, D)$. The latter would mean that $T(0 \,|\, c) \neq D(X_i^{\perp T} - c)$ for all $c \in \mathrm{supp}(T)$, which implies that $T$ is constant on $\mathrm{supp}(T)$. This again contradicts $\mathrm{supp}(T) \neq \{0\}$. $\qquad\square$

From now on whenever we assume that $D$ is a cutoff rule we suppose that $D$ does not depend on $I_{k,i}(c)$ with $c \neq 0$, that is $T$ and $D$ are synchronous regarding their cutoffs.

**Proposition 4.** *Let $D$ be a cutoff rule on $\mathbb{R}^K$ and let $i$ denote an individual. If $\dim(\mathrm{supp}(T)) = 1$ then $i \in \mathrm{At}(T, D) \cup \mathrm{Nt}(T, D) \cup \mathrm{DeF}(T, D) \cup \mathrm{ComP}(T, D)$.*

*Proof.* Since $\mathrm{supp}(T) \simeq \mathbb{R}$ one has $T_i = I_{k,i}$ with $k \in S(T)$. Suppose $i$ is in neither of the mentioned sets. Then $T_i(c) \neq D_i(c)$ and $T_i(\hat{c}) = D_i(\hat{c})$ for some $c, \hat{c} \in \mathrm{supp}(T)$. Thus, $\hat{c} = c + \lambda e_k$ for appropriate $\lambda \in \mathbb{R}$. If $T_i(c) = T_i(\hat{c})$ then $D_i(c) \neq D_i(\hat{c})$. Since $D$ is a cutoff rule and $I_{i,j}(c) = I_{i,j}(\hat{c})$ for $j \neq k$ we conclude $T_i(c) = I_{k,i}(c) \neq I_{k,i}(\hat{c}) = T_i(\hat{c})$. This contradicts the assumption. If otherwise $T_i(c) \neq T_i(\hat{c})$ one has $D_i(c) = D_i(\hat{c})$. Which would imply that $D_i(c)$ is constant for $c \in \mathrm{supp}(T)$, and thus $i \in \mathrm{At}(T, D) \cup \mathrm{Nt}(T, D)$. $\qquad\square$

## A.2 EXAMPLES

**Example (OR-Rules)** Let $D := \bigvee_{j=1}^{K} I_j$ and $T := \bigvee_{j=1}^{k} I_j$ for $k \in \{1, \ldots, K-1\}$. Then $\mathrm{supp}(T) = \mathbb{R}^k \times \{0\}^{K-k}$. Further, we have the following unit categorizations:

    1. $\mathrm{ComP}(T, D) = \{i \,|\, \forall k \leq j \leq K \,:\, X_{j,i} \leq 0\}$

        Note that in this case the additional or-conditions are zero, reducing the rule $D_i$ to $T_i$.

    2. $\mathrm{At}(T, D) = \{i \,|\, \exists k < j \leq K \,:\, X_{j,i} > 0\}$

        As long as there is any additional or condition that is always true, cutoff changes in $\mathrm{supp}(T)$ do not affect $D$.

    3. $\mathrm{Nt}(T, D) = \emptyset$

        Choose $c \in \mathrm{supp}(T)$ such that $X_{i,j} > c$ for $0 \leq j \leq k$, then $D_i = 1$.

    4. $\mathrm{DeF}(T, D) = \emptyset$

        Note that $T_i(c) = 1$ implies $D_i(c) = 1$ for all $c \in \mathrm{supp}(T)$.

**Example (XOR dominant rule)** Let $D := (I_1 \vee I_2) \wedge (\bar{I}_1 \vee \bar{I}_2)$ and $T := I_1$. Then $\mathrm{supp}(T) = \mathbb{R} \times \{0\}$. Note if $X_{2,i} > 0$, one has $D_i(c) = \bar{T}_i(c)$ and otherwise $D_i(c) = T_i(c)$ for all $c \in \mathrm{supp}(T)$. Thus:

1. $\mathrm{ComP}(T, D) = \{i \mid X_{2,i} \le 0\}$
2. $\mathrm{At}(T, D) = \emptyset$
3. $\mathrm{Nt}(T, D) = \emptyset$
4. $\mathrm{DeF}(T, D) = \{i \mid X_{2,i} > 0\}$

### A.3 EFFECT IDENTIFICATION

For this section, we require that the outcome does not directly depend on the treatment assignment $T$. Denote the set of all unit categories with

$$\mathcal{C} := \{\mathrm{ComP}(T, D), \mathrm{Nt}(T, D), \mathrm{At}(T, D), \mathrm{DeF}(T, D), \mathrm{Ind}(T, D)\}$$

and the set of non-change categories with:

$$\mathcal{C}^0 := \{\mathrm{Nt}(T, D), \mathrm{At}(T, D)\}$$

We assume that the categorization of a unit is independent of the support part of $T$ in a neighborhood of the cutoff, that is:

**Assumption 2.** *There exists some $\epsilon > 0$ such that*

$$\Pr(i \in \mathrm{Cat} \mid X_i^T = x) = \Pr(i \in \mathrm{Cat} \mid X_i^T = 0)$$

*for $\|x\| \le \epsilon$ and $\mathrm{Cat} \in \mathcal{C}$.*

Using this assumption and further assuming $\Pr\left(i \in \mathrm{Cat} \mid X_i^T = 0\right) > 0$ one has

$$\mathbb{E}\left(Y_i \mid X_i^T = x\right) = \sum_{\mathrm{Cat} \in \mathcal{C}} \mathbb{E}\left(Y_i \mid X_i^T = x, i \in \mathrm{Cat}\right) \Pr\left(i \in \mathrm{Cat} \mid X_i^T = 0\right)$$

and thus

$$\mathbb{E}\left(Y_i \mid X_i^T = x^+\right) - \mathbb{E}\left(Y_i \mid X_i^T = x^-\right) =$$
$$\sum_{\mathrm{Cat} \in \mathcal{C}} \left(\mathbb{E}\left(Y_i \mid X_i^T = x^+, i \in \mathrm{Cat}\right) - \mathbb{E}\left(Y_i \mid X_i^T = x^-, i \in \mathrm{Cat}\right)\right) \Pr\left(i \in \mathrm{Cat} \mid X_i^T = 0\right)$$

for appropriate directions $x^+, x^- \in \mathrm{supp}(T)$. Further, we employ a local continuity assumption, for the potential outcomes of the unit categories.

**Assumption 3.** *There exists an $\epsilon > 0$ such that $x \mapsto \mathbb{E}(Y_i(d) \mid X_i^T = x, i \in \mathrm{Cat})$ is continuous for $\|x\| \le \epsilon$, $d \in \{0, 1\}$ and $\mathrm{Cat} \in \mathcal{C}$.*

Note that

$$E\left(Y_i \mid X_i^T = x^\pm, i \in \mathrm{At}\right) = E\left(Y_i(1) \mid X_i^T = x^\pm, i \in \mathrm{At}\right)$$

and

$$E\left(Y_i \mid X_i^T = x^\pm, i \in \mathrm{Nt}\right) = E\left(Y_i(0) \mid X_i^T = x^\pm, i \in \mathrm{Nt}\right)$$

holds for the non-change unit categories. Together with Assumption 3 one derives:

$$\lim_{\lambda \to 0} \mathbb{E}\left(Y_i \mid X_i^T = \lambda x^+\right) - \lim_{\lambda \to 0} \mathbb{E}\left(Y_i \mid X_i^T = \lambda x^-\right) =$$
$$\sum_{\mathrm{Cat} \in \mathcal{C} \setminus \mathcal{C}^0} \left(\lim_{\lambda \to 0} \mathbb{E}\left(Y_i \mid X_i^T = \lambda x^+, i \in \mathrm{Cat}\right) - \lim_{\lambda \to 0} \mathbb{E}\left(Y_i \mid X_i^T = \lambda x^-, i \in \mathrm{Cat}\right)\right) \cdot \quad (3)$$
$$\cdot \Pr\left(i \in \mathrm{Cat} \mid X_i^T = 0\right)$$

Two more assumptions are in order to make further use of the continuity. First, we deny the existence of indecisive items, since this category does not separate the potential outcomes $Y_i(0)$ and $Y_i(1)$.

**Assumption 4.** $\mathrm{Ind}(T, D) = \emptyset$

Second, we assume that $x^+$ and $x^-$ induce a change in $T$.

**Assumption 5.**
$$1 = \lim_{\lambda \to 0} T\left(\lambda x^+ \,|\, 0\right) \neq \lim_{\lambda \to 0} T\left(\lambda x^- \,|\, 0\right) = 0$$

With this we know how $D_i$ behaves for complier and defier when approaching from $x^+$ and $x^-$ direction. That is:

$$\lim_{\lambda \to 0} \Pr\left(D_i = 1 \,|\, X_i^T = \lambda x^+, \, i \in \mathrm{ComP}\right) = 1 \text{ and } \lim_{\lambda \to 0} \Pr\left(D_i = 1 \,|\, X_i^T = \lambda x^-, \, i \in \mathrm{ComP}\right) = 0$$

as well as

$$\lim_{\lambda \to 0} \Pr\left(D_i = 1 \,|\, X_i^T = \lambda x^+, \, i \in \mathrm{DeF}\right) = 0 \text{ and } \lim_{\lambda \to 0} \Pr\left(D_i = 1 \,|\, X_i^T = \lambda x^-, \, i \in \mathrm{DeF}\right) = 1$$

Thus we can apply Assumption 3 to these two remaining categories on the right side of Equation 3 as well:

**Theorem 1.** *Let Assumptions 2, 3, 4 and 5 hold. Then the complier effect at the cutoff is identified as*

$$\mathbb{E}\left(Y_i(1) \,|\, X_i^T = 0, \, i \in \mathrm{ComP}\right) - \mathbb{E}\left(Y_i(0) \,|\, X_i^T = 0, \, i \in \mathrm{ComP}\right) =$$
$$\frac{1}{\Pr\left(i \in \mathrm{ComP} \,|\, X_i^T = 0\right)} \left(\lim_{\lambda \to 0} \mathbb{E}\left(Y_i \,|\, X_i^T = \lambda x^+\right) - \lim_{\lambda \to 0} \mathbb{E}\left(Y_i \,|\, X_i^T = \lambda x^-\right)\right) - C$$

*with $C$ being the correction term for defier:*

$$C := \frac{\Pr\left(i \in \mathrm{DeF} \,|\, X_i^T = 0\right)}{\Pr\left(i \in \mathrm{ComP} \,|\, X_i^T = 0\right)} \left(\mathbb{E}\left(Y_i(0) \,|\, X_i^T = 0, \, i \in \mathrm{DeF}\right) - \mathbb{E}\left(Y_i(1) \,|\, X_i^T = 0, \, i \in \mathrm{DeF}\right)\right)$$

We now investigate how dropping units in $\bigcup_{\mathrm{Cat} \in \mathcal{C}^0} \mathrm{Cat}$ affects the above identification result. For ease of presentation we make the assumption that there do not exist any defier, at the cutoff. Further, let $\Omega \subset \mathrm{At} \cup \mathrm{Nt}$. Then $\Pr\left(i \in \mathrm{ComP}, \, i \in \Omega \,|\, X_i^T = 0\right) = 0$ and thus one has

$$\Pr\left(i \in \mathrm{ComP} \,|\, X_i^T = 0\right) = \Pr\left(i \in \mathrm{ComP} \,|\, X_i^T = 0, \, i \notin \Omega\right) \Pr\left(i \notin \Omega \,|\, X_i^T = 0\right) \quad (4)$$

for the denominator. For the nominator note that

$$\mathbb{E}\left(Y_i \,|\, X_i^T = \lambda x^\pm, \, i \in \Omega\right) = \mathbb{E}\left(Y_i(0) \,|\, X_i^T = \lambda x^\pm, \, i \in \Omega \cap \mathrm{Nt}\right) \Pr\left(i \in \mathrm{Nt} \,|\, X_i^T = \lambda x^\pm, \, i \in \Omega\right)$$
$$+ \mathbb{E}\left(Y_i(1) \,|\, X_i^T = \lambda x^\pm, \, i \in \Omega \cap \mathrm{At}\right) \Pr\left(i \in \mathrm{At} \,|\, X_i^T = \lambda x^\pm, \, i \in \Omega\right)$$

holds. Requiring Assumption 3 and Assumption 2 to hold when conditioning on $\Omega \cap \mathrm{Nt}$ and $\Omega \cap \mathrm{At}$ (instead of $\mathrm{Nt}$ and $\mathrm{At}$) we get:

$$\lim_{\lambda \to 0} \mathbb{E}\left(Y_i \,|\, X_i^T = \lambda x^+, \, i \in \Omega\right) - \lim_{\lambda \to 0} \mathbb{E}\left(Y_i \,|\, X_i^T = \lambda x^-, \, i \in \Omega\right) = 0 \quad (5)$$

Since

$$\mathbb{E}\left(Y_i \,|\, X_i^T = \lambda x^\pm\right) = \mathbb{E}\left(Y_i \,|\, X_i^T = \lambda x^\pm, \, i \in \Omega\right) \Pr\left(i \in \Omega \,|\, X_i^T = \lambda x^\pm\right)$$
$$+ \mathbb{E}\left(Y_i \,|\, X_i^T = \lambda x^\pm, \, i \notin \Omega\right) \Pr\left(i \notin \Omega \,|\, X_i^T = \lambda x^\pm\right)$$

holds we require an assumption similar to Assumption 2 to for $\Pr\left(i \in \Omega \,|\, X_i^T = \lambda x^\pm\right)$ in order to get

$$\lim_{\lambda \to 0} \mathbb{E}\left(Y_i \,|\, X_i^T = \lambda x^+\right) - \lim_{\lambda \to 0} \mathbb{E}\left(Y_i \,|\, X_i^T = \lambda x^-\right)$$
$$= \left(\lim_{\lambda \to 0} \mathbb{E}\left(Y_i \,|\, X_i^T = \lambda x^+, \, i \notin \Omega\right) - \lim_{\lambda \to 0} \mathbb{E}\left(Y_i \,|\, X_i^T = \lambda x^-, \, i \notin \Omega\right)\right) \cdot$$
$$\cdot \Pr\left(i \notin \Omega \,|\, X_i^T = 0\right)$$
$$(6)$$

using Equation 5. Combining Equations 6 and 4 we obtain the following result:

**Theorem 2.** *Let Assumptions 2, 3, 4 and 5 hold, and $\Pr(i \in \mathrm{DeF} \,|\, X_i^T = 0) = 0$. Further, let $\Omega \subset \mathrm{Nt} \cup \mathrm{At}$ such that*

1. *there exists an $\epsilon > 0$ such that the functions $x \to \mathbb{E}\left(Y_i(0) \,\middle|\, X_i^T = x, \, i \in \Omega \cap \mathrm{Nt}\right)$ and $x \to \mathbb{E}\left(Y_i(1) \,\middle|\, X_i^T = x, \, i \in \Omega \cap \mathrm{At}\right)$ are continuous for $\|x\| < \epsilon$.*

2. *there exists an $\epsilon > 0$ such that*

$$\Pr\left(i \in \mathrm{Nt} \,\middle|\, X_i^T = x, \, i \in \Omega\right) = \Pr\left(i \in \mathrm{Nt} \,\middle|\, X_i^T = 0, \, i \in \Omega\right)$$

   *and*

$$\Pr\left(i \in \mathrm{At} \,\middle|\, X_i^T = x, \, i \in \Omega\right) = \Pr\left(i \in \mathrm{At} \,\middle|\, X_i^T = 0, \, i \in \Omega\right)$$

   *as well as*

$$\Pr\left(i \in \Omega \,\middle|\, X_i^T = x\right) = \Pr\left(i \in \Omega \,\middle|\, X_i^T = 0\right)$$

   *for $\|x\| < \epsilon$.*

*Then*

$$\mathbb{E}\left(Y_i(1) \,\middle|\, X_i^T = 0, \, i \in \mathrm{ComP}\right) - \mathbb{E}\left(Y_i(0) \,\middle|\, X_i^T = 0, \, i \in \mathrm{ComP}\right) =$$

$$\frac{1}{\Pr\left(i \in \mathrm{ComP} \,\middle|\, X_i^T = 0, \, i \notin \Omega\right)} \left( \lim_{\lambda \to 0} \mathbb{E}\left(Y_i \,\middle|\, X_i^T = \lambda x^+, \, i \notin \Omega\right) - \lim_{\lambda \to 0} \mathbb{E}\left(Y_i \,\middle|\, X_i^T = \lambda x^-, \, i \notin \Omega\right) \right)$$

*holds.*

## APPENDIX B  ADDITIONAL SIMULATION STUDY

In this appendix, we present an additional simulation study with more complex treatment assignments.

### B.1  DATA GENERATING PROCESS

We consider a multi-dimensional regression discontinuity design (MRD) with three scores $X_1, X_2, X_3$, where

$$(X_1, X_2, X_3)^\top \sim \mathcal{N}(0, \mathbf{I}_3).$$

For notational simplicity, we leave out the unit index $i$. Further, we generate independent covariates

$$Z_j \sim \mathrm{Uniform}(-1, 1)$$

for $j \in \{1, \ldots, d = 4\}$. The potential outcomes are defined as:

$$Y(0) = 0.1 \cdot \left( \sum_{i=1}^{3} X_i \right)^2 + g(Z) + \varepsilon$$

$$Y(1) = \tau - 0.4 \cdot \left( \sum_{i=1}^{3} X_i \right)^2 + a \cdot \left( \sum_{j=1}^{d} Z_j \right) \cdot \left( \sum_{i=1}^{3} X_i \right) + g(Z) + \varepsilon$$

where $\varepsilon \sim \mathcal{N}(0, 0.25)$, $\tau = 2$ is the treatment effect parameter at the joint cutoff point $(0, 0, 0)^\top$, and $a = 0.5$ controls the interaction between running variables and covariates. The function $g(Z)$ is defined as

$$g(Z) = \sum_{j=1}^{d} Z_j + \sum_{j=1}^{d} Z_j^2 + \sum_{1 \le j < k \le d} Z_j Z_k.$$

Define binary indicators for exceeding each cutoff as

$$I_k = \mathbb{1}[X_k > 0], \quad k = 1, 2, 3$$

where $c_1 = c_2 = c_3 = 0$ are the cutoff values for the scores. For the treatment assignment, we consider two settings:

- **Setting A**: $D_A = I_1 \wedge I_2 \wedge I_3$
- **Setting B**: $D_B = (I_1 \wedge I_2) \vee I_3$

The observed outcome is $Y = Y(0)(1 - D) + Y(1)D$.

## B.2 RDD ESTIMATES

The overall procedure consists of the following steps:

1. Choose a subrule $T$, which you would like to evaluate.

2. Based on $T$ and $D$, determine a subset of never- and alwaystakers $B(T, D) \subseteq \text{At}(T, D) \cup \text{Nt}(T, D)$, which can be identified.

3. Remove all observations from $B(T, D)$ from your original data.

4. Estimate a (fuzzy or sharp) univariate RDD on the remaining observations.

Remark that the subsets $B(T, D)$, will depend on the choice of $T$.
For simplicity, we will focus only on the evaluation of the first cutoff threshold, i.e. $T = I_1$.
As mentioned in Appendix D, one possibility to evaluate the effect would be to consider this as a fuzzy setting, where all observed units are used ("Fuzzy IV" approach). Instead, we would like to compare this approach to our proposal, which allows specifying exactly which units can be discarded, i.e. the "subset complier effect", which can be seen as a generalization of the "Frontier method".
In both settings, we will evaluate $T_1$ but with different choices of $D$. Considering the definition of $D_A$, there do not exist alwaystakers, but identifiable nevertakers, such that we can choose

$$B(T_1, D_A) := \{i | I_{2,i} = 0 \vee I_{3,i} = 0\} \subseteq \text{Nt}(T_1, D_A).$$

Instead in setting B, we can identify alwaystakers $\{i | I_{3,i} = 1\} \subseteq \text{At}(T_1, D_B)$ and nevertakers $\{i | I_{2,i} = 0 \wedge I_{3,i} = 0\} \subseteq \text{Nt}(T_1, D_B)$, such that we can choose

$$B(T_1, D_B) := \{i | I_{3,i} = 1\} \cup \{i | I_{2,i} = 0 \wedge I_{3,i} = 0\}.$$

In the following, we generate a dataset with 5,000 observations and evaluate the subsetting approach (*Subset*) against a fuzzy estimation approach on the whole dataset (*Full Dataset*). We estimate both approaches with the `rdrobust` (linear covariate adjustment) and `doubleml` (flexible covariate adjustment) packages, but without much tuning of the machine learning algorithms. The main focus of the comparison is the estimation on the full data as a fuzzy design and on the data subset as a sharp design.

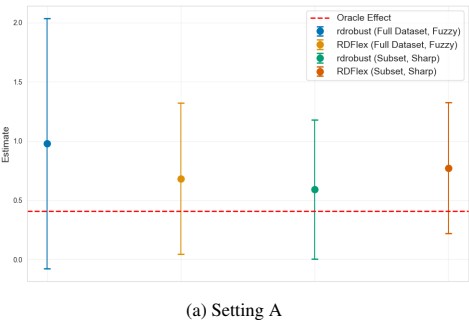
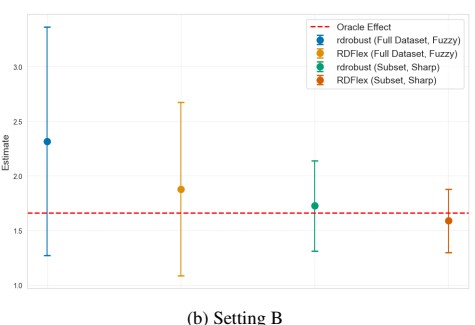

(a) Setting A                                       (b) Setting B

Figure 7: Comparison of point estimates and confidence intervals. For each method only using the data subset improves the precision of the estimator.

Note that the identified effect differs from $\tau$ since the potential outcomes depend on the scores. The oracle effect is computed on the subset of compliers (of $T_1$ with respect to $D$) using an independent sample of $100,000$ observations. It is obtained via kernel regression of the individual treatment effects $(Y(1) - Y(0))$ on the score $T_1$. Statistical coverage is evaluated over 200 independent datasets. The confidence interval length for the subset methods is substantially smaller, while still maintaining the desired coverage level.

## APPENDIX C ADDITIONAL NUMERICAL RESULTS

In this appendix, we present additional results concerning Section 5.1 and 5.2.

Table 2: Simulation Results Setting A

| Data | Method | Mean Bias | s.e. | Mean CI Length | Coverage |
|------|--------|-----------|------|----------------|----------|
| Full Dataset | rdrobust (Fuzzy) | -0.012 | 0.483 | 1.816 | 0.945 |
| | RDFlex (Fuzzy) | 0.020 | 0.351 | 1.253 | 0.930 |
| Subset | rdrobust (Sharp) | -0.039 | 0.298 | 1.108 | 0.940 |
| | RDFlex (Sharp) | -0.026 | 0.285 | 1.025 | 0.945 |

Table 3: Simulation Results Setting B

| Data | Method | Mean Bias | s.e. | Mean CI Length | Coverage |
|------|--------|-----------|------|----------------|----------|
| Full Dataset | rdrobust (Fuzzy) | 0.077 | 0.568 | 2.148 | 0.960 |
| | RDFlex (Fuzzy) | 0.042 | 0.470 | 1.821 | 0.955 |
| Subset | rdrobust (Sharp) | 0.033 | 0.223 | 0.837 | 0.925 |
| | RDFlex (Sharp) | 0.008 | 0.172 | 0.633 | 0.940 |

## C.1 TABLE OF RDD ESTIMATES AT $I_D$ AND $I_Y$

Tables 4 and 5 provide additional insights to the estimation conducted in Section 5.1. Particularly, the quality of fit in the ML estimation, the coverage as well as the mean bias can be assessed here.

| setting | method | Mean Bias | s.e. | Coverage | RMSE left | Log loss left | RMSE right | Log loss right |
|---------|--------|-----------|------|----------|-----------|---------------|------------|----------------|
| Fuzzy | RDD Conventional Covs | -0.0025 | 0.0111 | 0.9912 | | | | |
| | RDD Without Covs | -0.0027 | 0.0109 | 0.9735 | | | | |
| | RDFlex Lasso | -0.0030 | 0.0092 | 0.9823 | 0.0242 | 0.0787 | 0.0897 | 0.5645 |
| | RDFlex Stacking | -0.0029 | 0.0092 | 0.9823 | 0.0243 | 0.0169 | 0.0898 | 0.4938 |
| Fuzzy Subset | RDD Conventional Covs | -0.0001 | 0.0117 | 0.9609 | | | | |
| | RDD Without Covs | -0.0003 | 0.0118 | 0.9478 | | | | |
| | RDFlex Lasso | -0.0002 | 0.0110 | 0.9435 | 0.0249 | 0.4230 | 0.1010 | 0.5140 |
| | RDFlex Stacking | -0.0004 | 0.0113 | 0.9522 | 0.0258 | 0.0329 | 0.1011 | 0.3580 |
| Sharp | RDD Conventional Covs | -0.0040 | 0.0063 | 0.9120 | | | | |
| | RDD Without Covs | -0.0040 | 0.0063 | 0.9240 | | | | |
| | RDFlex Lasso | -0.0040 | 0.0055 | 0.9200 | 0.0242 | | 0.0886 | |
| | RDFlex Stacking | -0.0040 | 0.0055 | 0.9240 | 0.0243 | | 0.0888 | |
| Sharp Subset | RDD Conventional Covs | -0.0014 | 0.0094 | 0.9680 | | | | |
| | RDD Without Covs | -0.0014 | 0.0095 | 0.9600 | | | | |
| | RDFlex Lasso | -0.0014 | 0.0079 | 0.9560 | 0.0246 | | 0.1004 | |
| | RDFlex Stacking | -0.0015 | 0.0080 | 0.9600 | 0.0256 | | 0.1006 | |

Table 4: Mean bias, standard errors, coverage, and quality of fit for the different estimators at $c_D$ in the setting of Section 5.1.

## C.2 VALIDATION TESTS FOR THE REAL DATA

A typical validation for the application of RDD is the pseudo-cutoff test (Cattaneo et al., 2019). We compare the estimated intent to treat effect of the real data at $c_D$ with a fictional lower and a higher cutoff. In Table C.2 it is visible that both pseudo cutoffs show effects close to zero and with large confidence band, while at the real cutoff the effect is almost significant on $95\%$ confidence level.

| Cutoff | Coef | 2.5 % CI | 97.5 % CI |
|--------|------|----------|-----------|
| lower pseudo | 0.002919 | -0.014796 | 0.020634 |
| real | 0.039332 | -0.001524 | 0.080188 |
| higher pseudo | 0.041137 | -0.015333 | 0.097606 |

Table 6: Coefficient and confidence interval of the sharp real data estimation at pseudo cutoffs concerning $c_D$.

| setting | method | Mean Bias | s.e. | Coverage | RMSE left | Log loss left | RMSE right | Log loss right |
|---|---|---|---|---|---|---|---|---|
| Fuzzy | RDD Conventional Covs | -0.0158 | 0.0124 | 0.6867 | | | | |
| | RDD Without Covs | -0.0189 | 0.0117 | 0.5200 | | | | |
| | RDFlex Lasso | -13.0991 | 690.4369 | 0.9600 | 0.0275 | 0.0202 | 0.0707 | 0.4681 |
| | RDFlex Stacking | -0.3166 | 16.4743 | 0.9667 | 0.0257 | 0.0104 | 0.0732 | 0.3574 |
| Fuzzy Subset | RDD Conventional Covs | -0.0003 | 0.0085 | 0.9754 | | | | |
| | RDD Without Covs | -0.0002 | 0.0085 | 0.9713 | | | | |
| | RDFlex Lasso | 0.0002 | 0.0057 | 0.9795 | 0.0333 | 0.2688 | 0.0936 | 0.1580 |
| | RDFlex Stacking | 0.0000 | 0.0058 | 0.9754 | 0.0337 | 0.0321 | 0.0938 | 0.1460 |
| Sharp | RDD Conventional Covs | -0.0043 | 0.0073 | 0.8720 | | | | |
| | RDD Without Covs | -0.0050 | 0.0074 | 0.8880 | | | | |
| | RDFlex Lasso | -0.0045 | 0.0077 | 0.8800 | 0.0271 | | 0.0726 | |
| | RDFlex Stacking | -0.0048 | 0.0070 | 0.8680 | 0.0255 | | 0.0732 | |
| Sharp Subset | RDD Conventional Covs | -0.0005 | 0.0078 | 0.9800 | | | | |
| | RDD Without Covs | -0.0004 | 0.0078 | 0.9720 | | | | |
| | RDFlex Lasso | -0.0003 | 0.0054 | 0.9760 | 0.0333 | | 0.0940 | |
| | RDFlex Stacking | -0.0004 | 0.0054 | 0.9720 | 0.0337 | | 0.0939 | |

Table 5: Mean bias, standard errors, coverage, and quality of fit for the different estimators at $c_Y$ in the setting of Section 5.1.

## C.3 SHARP TWO-DIMENSIONAL DESIGN

We provide additional results for Section 5.1 in a different DGP configuration, where we assume that the operator shows perfect compliance with the decision taken by the decision tool ($D = T$). This yields a perfect sharp two-dimensional RDD.

### C.3.1 DISTANCE CUT-OFF

The effect at $I_D$ is negative and significant (See Figure 8 and Table 7). While the setting in Section 5.1 yielded $D \neq T$ for units where the operator is cautious about the rework, here all units are reworked independent of additional information that hint at a possibly negative outcome of the rework. Thus, the effect of the rework is even more negative and the proposed movement of the threshold should be larger than in the setting above.

There is no fuzzy subset estimator, as the conditioned set on the compliers of $I_D$ is sharp only. The subset estimator again reduces bias at a lightly higher standard error in the intent-to-treat (sharp) estimator. The true sharp subset effect is smaller as less nevertakers are included that have an effect of zero and thus take the average effect closer to zero. The ML adjustment reduces the standard error for all estimators.

| setting | method | Mean Bias | s.e. | Coverage | RMSE left | Log loss left | RMSE right | Log loss right |
|---|---|---|---|---|---|---|---|---|
| Fuzzy | RDD Conventional Covs | -0.0019 | 0.0102 | 0.4400 | | | | |
| | RDD Without Covs | -0.0017 | 0.0103 | 0.4440 | | | | |
| | RDFlex Lasso | -0.0021 | 0.0083 | 0.4440 | 0.0242 | 0.1130 | 0.0958 | 0.4996 |
| | RDFlex Stacking | -0.0018 | 0.0085 | 0.4440 | 0.0243 | 0.0135 | 0.0960 | 0.3813 |
| Sharp | RDD Conventional Covs | -0.0042 | 0.0067 | 0.9320 | | | | |
| | RDD Without Covs | -0.0042 | 0.0067 | 0.9360 | | | | |
| | RDFlex Lasso | -0.0042 | 0.0057 | 0.9360 | 0.0241 | | 0.0951 | |
| | RDFlex Stacking | -0.0042 | 0.0057 | 0.9320 | 0.0243 | | 0.0954 | |
| Sharp Subset | RDD Conventional Covs | 0.0010 | 0.0101 | 0.9720 | | | | |
| | RDD Without Covs | 0.0009 | 0.0102 | 0.9720 | | | | |
| | RDFlex Lasso | 0.0009 | 0.0085 | 0.9720 | 0.0247 | | 0.1078 | |
| | RDFlex Stacking | 0.0009 | 0.0085 | 0.9720 | 0.0258 | | 0.1080 | |

Table 7: Mean bias, standard errors, coverage, and quality of fit for the different estimators at $I_D$ in the setting without nevertakers (sharp MRD).

### C.3.2 YIELD CUT-OFF

The effect at $I_Y$ is comparable to the DGP setting in Section 5.1 (See Figure 9 and Table 8).

## C.4 NOISELESS DGP

We provide additional results for Section 5.1 in a setting where we assume that there is less noise in the system. This is to provide a benchmark where the true ground truth is less noisy. We only

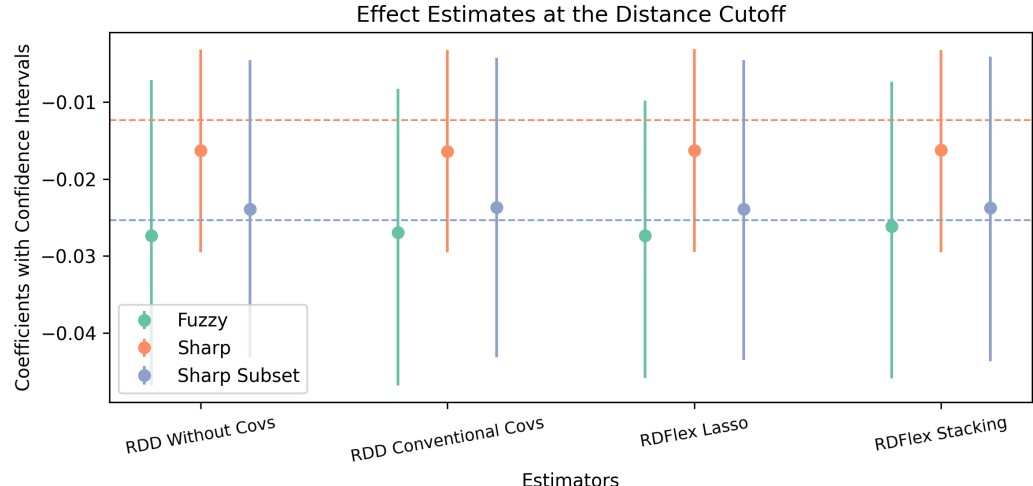

Figure 8: Median coefficient and median $95\%$ confidence interval for coefficients at $I_D$. The estimation was repeated $r = 250$ in the simulated process with a sharp boundary ($D = T$). The different colors depict a fuzzy, a sharp, and a sharp subset estimator. The dashed line shows the oracle estimate for each setting, which overlap for some settings.

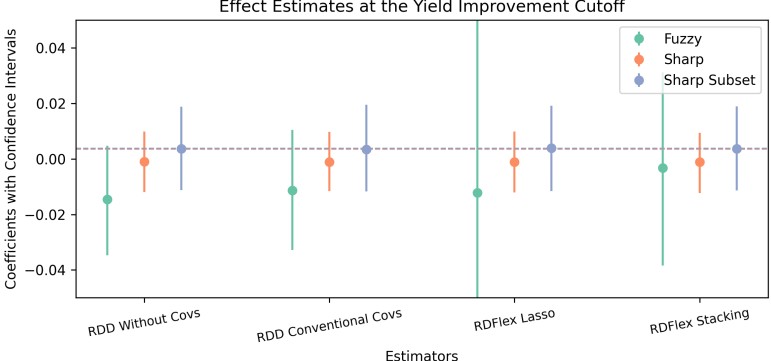

Figure 9: Median coefficient and median $95\%$ confidence interval for coefficients at $I_Y$. The estimation was repeated $r = 250$ in the simulated process with a sharp boundary ($D = T$). The different colors depict a fuzzy, a sharp, and a sharp subset estimator. The dashed line shows the oracle estimate for each setting, which overlap for some settings.

evaluate $I_D$ as due to little variation around $I_Y$ and the design of the process the estimators for $I_Y$ have large variance.

### C.4.1 DISTANCE CUT-OFF

In this setting, all estimates are positive (See Figure 10 and Table 9). This could hint that the negative effect in Section 5.1, which does not match with our estimation in the real data in Section 5.2, can be explained by the tuning of the noise parameters of the process. The coverage is held better on the subset estimators in this DGP setting. The sharp subset effect again is larger due to less nevertakers.

| setting | method | Mean Bias | s.e. | Coverage | RMSE left | Log loss left | RMSE right | Log loss right |
|---|---|---|---|---|---|---|---|---|
| Fuzzy | RDD Conventional Covs | -0.0160 | 0.0125 | 0.3760 | | | | |
| | RDD Without Covs | -0.0193 | 0.0115 | 0.3040 | | | | |
| | RDFlex Lasso | -0.0697 | 4.5786 | 0.5920 | 0.0276 | 0.0213 | 0.0744 | 0.4486 |
| | RDFlex Stacking | 0.0001 | 1.7755 | 0.5760 | 0.0257 | 0.0096 | 0.0777 | 0.3278 |
| Sharp | RDD Conventional Covs | -0.0035 | 0.0081 | 0.8480 | | | | |
| | RDD Without Covs | -0.0044 | 0.0079 | 0.8600 | | | | |
| | RDFlex Lasso | -0.0041 | 0.0086 | 0.8680 | 0.0271 | | 0.0766 | |
| | RDFlex Stacking | -0.0042 | 0.0077 | 0.8600 | 0.0256 | | 0.0771 | |
| Sharp Subset | RDD Conventional Covs | -0.0008 | 0.0081 | 0.9760 | | | | |
| | RDD Without Covs | -0.0008 | 0.0080 | 0.9760 | | | | |
| | RDFlex Lasso | -0.0006 | 0.0056 | 0.9800 | 0.0336 | | 0.0971 | |
| | RDFlex Stacking | -0.0007 | 0.0055 | 0.9760 | 0.0333 | | 0.0970 | |

Table 8: Mean bias, standard errors, coverage, and quality of fit for the different estimators at $I_Y$ in the setting without nevertakers (sharp MRD).

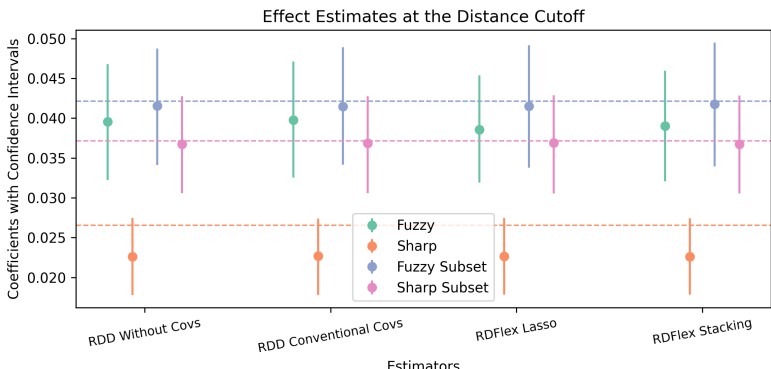

Figure 10: Median coefficient and median $95\%$ confidence interval for coefficients at $_D$. The estimation was repeated $r = 250$ in the simulated process with no noise. The different colors depict a fuzzy, a sharp, and a fuzzy and sharp subset estimator. The dashed line shows the oracle estimate for each setting, which overlap for some settings.

# APPENDIX D COMMON ESTIMATORS FOR MULTI-SCORE REGRESSION DISCONTINUITY

The following estimators for multi-score RD designs (e.g. consisting of two score components $\{X_1, X_2\}$) are commonly proposed in recent surveys by Wong et al. (2013), Porter et al. (2017), or Reardon & Robinson (2012)).

| setting | method | Mean Bias | s.e. | Coverage | RMSE left | Log loss left | RMSE right | Log loss right |
|---|---|---|---|---|---|---|---|---|
| Fuzzy | RDD Conventional Covs | -0.0021 | 0.0038 | 0.9040 | | | | |
| | RDD Without Covs | -0.0022 | 0.0038 | 0.8920 | | | | |
| | RDFlex Lasso | -0.0034 | 0.0032 | 0.8840 | 0.0240 | 0.0937 | 0.0256 | 0.5585 |
| | RDFlex Stacking | -0.0030 | 0.0032 | 0.9080 | 0.0237 | 0.0113 | 0.0247 | 0.4681 |
| Fuzzy Subset | RDD Conventional Covs | -0.0006 | 0.0038 | 0.9600 | | | | |
| | RDD Without Covs | -0.0006 | 0.0039 | 0.9680 | | | | |
| | RDFlex Lasso | -0.0007 | 0.0033 | 0.9560 | 0.0240 | 0.4183 | 0.0120 | 0.4649 |
| | RDFlex Stacking | -0.0006 | 0.0034 | 0.9640 | 0.0248 | 0.0301 | 0.0129 | 0.2750 |
| Sharp | RDD Conventional Covs | -0.0038 | 0.0025 | 0.5920 | | | | |
| | RDD Without Covs | -0.0039 | 0.0025 | 0.6040 | | | | |
| | RDFlex Lasso | -0.0039 | 0.0021 | 0.5960 | 0.0240 | | 0.0254 | |
| | RDFlex Stacking | -0.0038 | 0.0021 | 0.6080 | 0.0237 | | 0.0246 | |
| Sharp Subset | RDD Conventional Covs | -0.0006 | 0.0032 | 0.9680 | | | | |
| | RDD Without Covs | -0.0006 | 0.0032 | 0.9720 | | | | |
| | RDFlex Lasso | -0.0006 | 0.0027 | 0.9760 | 0.0240 | | 0.0121 | |
| | RDFlex Stacking | -0.0006 | 0.0027 | 0.9720 | 0.0248 | | 0.0128 | |

Table 9: Mean bias, standard errors, coverage, and quality of fit for the different estimators at $I_D$ in the setting without noise in the DGP.

1. **Binding the score**: This method combines both score components into one dimension by aggregating the minimum or the maximum of both scores. In settings where there is a uniform effect along the boundary, this technique makes it possible to estimate the effect of one treatment level with standard RDD estimators, using the score variable $x^* = \min(X_1, X_2)$ around a normalized cut-off.

2. **Frontier method**: This method divides the two-dimensional setting into two one-dimensional ones. It allows to evaluate two separate RDD along cutoff $c_1$ and $c_2$, respectively. Non-compliers due to the other dimension of $X$ are discarded.

3. **Location Specific Effect**: Similarly to the latter two methods that used a $L_1$ distance from the boundary to transform the two-dimensional setting into an easier-to-estimate one-dimensional one, it is also possible to use the $L_2$-distance of each observation from a specific location at the boundary. This is particularly popular in the geographic RDD literature since we are able to identify location specific effects $\tau(X_1, X_2)$ instead of averaged effects along one dimension.

4. **Fuzzy IV**: This method divides the two-dimensional setting into two one-dimensional ones similarly to (2). However, the method does not discard the observations that do not comply to the derived one-dimensional rule. The position relative to the cutoff is now only an IV for receiving the treatment, there is "non-compliance" caused by treatment assignment based on the other dimension.

5. **Parametric Surface**: This method is the only setting where the two-dimensional structure of a RDD with two scores is directly taken into account. Using a parametric or nonparametric approach, the two-dimensional surface of treatment and control groups are estimated and the effect can be estimated as the average difference in outcomes at the boundary.

## Appendix E  DGP for the cautious operator

As discussed in Section 4 we assume an information advantage of the final decision maker $D$. To model this algorithmically we suppose that $T$ has only access to the improvement estimates of every $m$-th item in the production lot resulting in the yield score $X_Y$ whereas the final decision maker knows the estimate for every item $X_E$. In particular, the overall estimated yield improvement is $X_E = X_Y + X_R$ where $X_R$ is the yield improvement of items not taken into account by $X_Y$. In the notation of Section 4 we set $X_{op} := X_E$. That is, the operator specific policy in Algorithm 1 outlined below is $D_O(X_Y, X_E) := I_{X_Y} \wedge I_{X_E}$.

---

**Algorithm 1:** DGP for a cautious operator

---

**Data:** seed, lot-size $n$, cutoff $c$, measurement steps $m$, yield criteria $\mathcal{Y}$, distance criteria $\mathcal{D}$, operator specific policy $D_O$

**Result:** lot $L$, scores $X = (X_D, X_Y)$, assigned treatment $T$, actual treatment $D$, outcome $Y$

$L \leftarrow (C_1, \ldots, C_n)$ generate a random production lot of $n$ items;

$X_D \leftarrow \mathcal{D}(L)$ calculate the distance to the target;

$\hat{L}_A \leftarrow$ carry out an optimal rework step on $L$;

$X_Y \leftarrow$ calculate improvement $\mathcal{Y}(\hat{L}_A) - \mathcal{Y}(L)$ on every $m$-th item;

$T \leftarrow \mathbb{1}[X_D > c_D] \wedge \mathbb{1}[X_Y > c_Y]$;

$X_E \leftarrow$ calculate improvement $\mathcal{Y}(\hat{L}_A) - \mathcal{Y}(L)$ on every item;

$D \leftarrow \mathbb{1}[X_D > c_D] \wedge D_O(X_Y, X_E)$;

$L_A \leftarrow$ carry out a realistic (noisy) rework step on $L$;

**if** $D$ **then**

   |   $Y \leftarrow \mathcal{Y}(L_A)$

**else**

   |   $Y \leftarrow \mathcal{Y}(L)$

**end**

---

## APPENDIX F  COMPUTATIONAL RESOURCES AND HYPERPARAMETERS

. In this appendix, we specify the computational resources that were used to achieve the main results.

The simulated study in Section 5.1 was facilitated on a single node of a high performance cluster. Specifically, 16 cores of an AMD EPYC 9654 CPU with 8GB RAM were used. The total computation time was 9.5 hours for 250 repetitions with 4 estimators in 8 scenarios (each 4 scenarios concerning $I_D$ and $I_Y$).

The real data application in Section 5.2 was facilitated on single node spark cluster. Specifically, 4 cores of a 64bit Intel(R) Xeon(R) Platinum 8171M CPU with 8 GB RAM were used. The total computation time was 2 hours.

For the no-covariate and conventional-covariate RDD estimation, the `rdrobust` (Calonico et al., 2017) package was used with default parameters regarding bandwidth selection and polynomial order. For the flexible covariate adjustment, the `DoubleML` (Bach et al., 2022) package was used with `fit_iterations = 2` and 5-fold cross-fitting. For ML, the `scikit-learn` (Pedregosa et al., 2011) implementation of cross-validated Lasso was used, and the stacked learner was defined as follows: `stacking_regressor = StackingRegressor( estimators=[ ('lgbm_regressor', LGBMRegressor(n_estimators=100, learning_rate=0.01, verbose=-1, n_jobs=-1)), ('global_forest', GlobalRegressor(RandomForestRegressor(n_jobs=-1))), ('linear_regressor', LassoCV(n_jobs=-1)), ], final_estimator=RidgeCV(), n_jobs=-1).`

## APPENDIX G  STATEMENT ON AI USAGE

For this research paper, large language models were used solely to assist with literature search, writing, and coding. All conceptualization, ideation, and theoretical contributions were carried out without AI support. The paper and code were authored entirely by the researchers, with AI serving only as a support and feedback tool.

