# OpenReview forum: "Bringing Light to the Threshold: Identification of Multi-Score Regression Discontinuity Effects with Application to LED Manufacturing"
_ICLR.cc/2026/Conference — Submitted to ICLR 2026_

### Official Review · Reviewer_Wgrp · 2025-10-25

**Soundness:** 2
**Presentation:** 2
**Contribution:** 2
**Rating:** 4
**Confidence:** 2

**Summary:**

The paper extends the classic Regression Discontinuity Design (RDD) framework to multi-score RDD (MRD), where treatment decisions are based on multiple thresholds. This extension is particularly useful in operational contexts like manufacturing, where decisions are made based on more complex rules rather than a single score. The paper presents a novel identification result for the complier effect in multi-dimensional cutoff settings. Through unit categorization(compliers, defiers, always takers, never takers, and indecisive units), the authors develop a rigorous framework for analyzing treatment effects in such contexts. The theoretical contributions are complemented by an empirical applicationto LED production data, demonstrating the practical utility of MRD in optimizing production policies and decision-making.

**Strengths:**

The proposed identification result for the complier effect in MRD is valuable, especially for complex decision-making systems like those found in industrial operations. By expanding the RDD framework to handle multi-dimensional decision rules, the paper contributes to making causal inference applicable to more realistic settings.

Unit categorization (compliers, defiers, always takers, never takers) is a useful framework for understanding how different units respond to treatment in multi-score settings. The inclusion of indecisive units is an important and novel contribution to the literature.

The empirical application to LED production shows how the MRD framework can be applied in a real-world manufacturing context. The results provide insights into how production policies could be adjusted to improve efficiency, demonstrating the practical applicability of the MRD framework.

**Weaknesses:**

The exposition of the paper is somewhat unclear, especially given the many definitions, lemmas, and propositions. These technical details are not always well-motivated or explained. A more intuitive approach would be to use the empirical application as a running example throughout the theoretical section, which would make the theoretical results more tangible. Since the empirical data is 2-dimensional, I suggest that the authors limit the theoretical results to the 2-dimensional case and move the more general case to the appendix.

The unit categorization section introduces multiple propositions and a lot of notation to formalize the different types of units (compliers, defiers, etc.). However, this section is not well connected to the empirical relevance of the results. Although Theorem 1 relies on unit categorization, the paper doesn’t clearly explain how these concepts apply to the data or how they influence the interpretation of the results. It might be more efficient to introduce these categories in a more intuitive way, rather than getting bogged down in formalism early on. The reader would benefit from a clearer link between the theoretical framework and the empirical application.

The discussion of assumptions related to the continuity and unit categorization in MRD is valuable but would benefit from further elaboration on how these assumptions might hold or fail in different empirical settings, particularly for industrial applications.

While the empirical section demonstrates the applicability of the framework, additional robustness checks on the sensitivity of results to changes in thresholds or decision rules would provide stronger evidence for the framework's validity in production contexts.

**Questions:**

Have you considered including visualizations to help illustrate the unit categorization and the identification results? In the RDD literature, visual aids, such as plots of discontinuities, are a valuable tool for empirical researchers. For example, Figure 4 is a great example of how to present the results visually, which aids in understanding the treatment effect at the cutoff.

While the paper focuses on identification, it would be helpful to provide more details on how researchers can implement the estimation procedure based on the identification results. For instance, how should researchers practically apply the unit categorization framework to estimate treatment effects using the MRD method? Providing a more concrete roadmap for applying the results would improve the paper’s practical value, making the theoretical insights easier to translate into real-world applications.

---

> ### Author Response · Authors · 2025-11-25
>
> Thanks a lot for your time and efforts in reviewing our paper. We carefully read your feedback and provide a revised version to address your questions and concerns.
>
> Concerning your questions:
>
> *Have you considered including visualizations to help illustrate the unit categorization and the identification results? In the RDD literature, visual aids, such as plots of discontinuities, are a valuable tool for empirical researchers. For example, Figure 4 is a great example of how to present the results visually, which aids in understanding the treatment effect at the cutoff.*
>
> We fully agree that visuals are an important tool in RDD, in particular in the univariate case. In our multi-dimensional case, visualizations become a bit more challenging.  We have included conceptional plots in Figures 1 and 2 in the revised version (Section 3), and hope they shed more light on the unit categorizations and particularly the indecisive units.
>
> *While the paper focuses on identification, it would be helpful to provide more details on how researchers can implement the estimation procedure based on the identification results. For instance, how should researchers practically apply the unit categorization framework to estimate treatment effects using the MRD method? Providing a more concrete roadmap for applying the results would improve the paper’s practical value, making the theoretical insights easier to translate into real-world applications.*
>
> You are completely right that the focus of this paper (and the results) are on identification. Our results give guidance on which kind of treatment effects are identified in MRDs. Of course estimation of the identified effect is then also key.
>
> As suggested by Reviewer MKKS, we have added a simulation study to the appendix (see Appendix B (page 18) in revised version), which considers a more complex composition of decision rules. In this example, we explain how to subset and categorize units and include a short list of the most important steps. This highlights the actual implementation and provides a more concrete example and roadmap on how to apply our results.

---

### Official Review · Reviewer_MKKS · 2025-10-31

**Soundness:** 2
**Presentation:** 2
**Contribution:** 3
**Rating:** 4
**Confidence:** 4

**Summary:**

This paper proposes a generalized framework for the multi-score regression discontinuity (MRD) design, extending the classic RDD to scenarios with multiple cutoffs. In particular, it allows for arbitrary combinations of Boolean operations and further formalizes the classification of units and the identification of the compiler effect. Furthermore, the paper conducts empirical studies on semi-synthetic and real-world LED semiconductor manufacturing data.

**Strengths:**

1. The paper provides a generalized and formalized theoretical framework for MRD. Compared to traditional methods that often consider only special cases, the paper's framework accommodates multidimensional cutoffs under general Boolean-type threshold rules. It also introduces the concept of indecisive units, which do not exist in the single-dimensional RDD setting.

2. The paper conducts an empirical analysis on real-world data from opto-electronic/LED manufacturing, highlighting the practical significance of the MRD framework.

3. The paper systematically tests and compares various ML estimators within its theoretical framework, providing valuable guidance for the practical application of MRD.

**Weaknesses:**

1. The experiments are conducted only under a binary "AND" rule. Given the paper's core contribution, it would benefit from experimental demonstrations under higher-dimensional and more complex combinations of Boolean rules, even if real-world datasets for such scenarios are unavailable.

2. The experiments lack a comparison with other existing MRD methods (as mentioned in Appendix C). Such a comparison is important for demonstrating the advantages of the proposed method, even in the two-dimensional scenario.

3. The paper's emphasis seems limited to its value in industrial applications. A lack of discussion on other potential MRD scenarios (e.g., in healthcare or economics) might limit its perceived general applicability.

4. The notation used may lack sufficient explanation for readers who are not familiar with the potential outcomes framework.

**Questions:**

1. Could the authors experimentally demonstrate the estimation performance under higher-dimensional and more complex Boolean rule combinations?

2. Is it possible to provide a comparison with other existing MRD methods within the current two-dimensional setting?

3. Could the concept of indecisive units be discussed in greater depth? This discussion would be particularly valuable in the context of the practical application setting, as well as regarding the impact of violating Assumption 4 (which assumes their absence) on the proposed estimators.

---

> ### Author Response · Authors · 2025-11-25
>
> Thanks a lot for your time and efforts in reviewing our paper. We carefully read your feedback and provide a revised version to address your questions and concerns.
>
> Concerning your questions:
>
> *Could the authors experimentally demonstrate the estimation performance under higher-dimensional and more complex Boolean rule combinations?*
>
> We have added an additional simulation to the appendix, where we evaluate a part of a decision rule which consists of three rule combinations. The results generally show better precision for the subsetting approach, while maintaining the same coverage properties (with linear or flexible covariate adjustments). Please refer to Appendix B (page 18) in the revised version.
>
> *Is it possible to provide a comparison with other existing MRD methods within the current two-dimensional setting?*
>
> The focus of the paper and the main results are on identification of causal effects in the MRD, mainly **along a single score** to allow for straightforward improvements of the decision rule. Compared to other MRD methods this identifies a different effect as, e.g., border-location specific effects in a geographical setting or an approach that is binding the scores into a single component. Such a comparison would then not be very informative and could be confusing for readers as the target of each method is a different type of intervention.
>
> *Could the concept of indecisive units be discussed in greater depth? This discussion would be particularly valuable in the context of the practical application setting, as well as regarding the impact of violating Assumption 4 (which assumes their absence) on the proposed estimators.*
>
> Thanks for this important point. To improve the understandability of our unit categorization we have added two figures in Section 3 (Figure 1 and 2 in the revised version) that also address a similar point raised by Reviewer Wgrp.
>
> In our application the final decision $D$ is based on the human operator, whereas $T$ refers to the automatic decision rule based on the distance and yield improvement scores. An indecisive item (of $T$ with respect to $D$) would be a chip lot, where the operator does not always comply with the automatic decision rule and does not give a constant response, i.e. no nevertaker or alwaystaker. In the scope of the new figure in the revised version, this means the final decision $D$ can differ in an unstructured way from $T$ for identical additional information $N^T$ at different cutoff thresholds. Remark, that the absence of indecisive items only has to hold locally around the cutoff. We assume that the operator does not change the override behavior $D$, and thus the inclination to follow the rule $T$, if we only slightly move the cutoffs. This assumption is on par with the usual absence of defiers assumptions.
>
> Specifically the cautious operator, who avoids degradation via unnecessary rework steps via additional additional information, would exclude indecisive items, because his behavior is constant in the scope of the observed rules, i.e. the operator only decides which unit is a nevertaker.

---

### Official Review · Reviewer_rcgL · 2025-11-03

**Soundness:** 4
**Presentation:** 4
**Contribution:** 4
**Rating:** 8
**Confidence:** 3

**Summary:**

Reasonable extension of RDD with novel real world data application

**Strengths:**

- This paper stands out due to its honesty about the strength of its assumptions: Line 483 onwards:“Causal identification is not possible without any of these assumptions. The question is whether such an assumption is interpretable enough to justify an approximate conformance in empirical studies. Compared to previously presented formulations, we draw from the intuitive language of unit categories to aid the argument for or against applicability.”
- It has a clear problem statement and provides reasonable extensions of established RDD methods as a possible solution.

**Weaknesses:**

- See questions

**Questions:**

- Line 037: can you provide a definition for ‘score’ here already?  Even informal intuition to increase accessibility might be good
- Line 039 “When correctly specified”, can you briefly summarise what’s required for correct ‘specification’? There is no free lunch, so if the method can identify without assuming uncofundness or positivity, it usually needs to assume something else.
- Line 117: What does ‘credible’ mean here exactly? Is it a formal term in Statistics?
- Line470: You suggest that opposite signs can be “can be explained by the calibration of the semi-synthetic process.” Can you provide additional theories apart from calibration that could yield such a result?

---

> ### Author Response · Authors · 2025-11-25
>
> Thanks a lot for your time and efforts in reviewing our paper. We really appreciated your feedback and also that you stress the novelty!
>
> Concerning your questions:
>
> *Line 037: can you provide a definition for ‘score’ here already? Even informal intuition to increase accessibility might be good*
>
> Thanks for this point! We have added examples to increase accessibility:
>
> > Typical examples are credit scores, GPAs or vote shares.
>
> *Line 039 “When correctly specified”, can you briefly summarise what’s required for correct ‘specification’? There is no free lunch, so if the method can identify without assuming uncofundness or positivity, it usually needs to assume something else.*
>
>
> This refers to the exact definitions specified in 3.3. Typically, the main assumptions require continuity of the potential outcomes and no manipulation of the score variable.
>
> *Line 117: What does ‘credible’ mean here exactly? Is it a formal term in Statistics?*
>
> In this context credible is not a formal term and just used for comparison with randomized experiments.
>
> *Line 470: You suggest that opposite signs can be “can be explained by the calibration of the semi-synthetic process.” Can you provide additional theories apart from calibration that could yield such a result?*
>
> The opposite sign could mainly be explained by the imperfect modeling of the operator decision, which only takes incomplete information into account. Particularly, the real-world operator might have more knowledge, such as machine states and outputs, as well as practical job experience.
>
> Thanks again for your review!

---

> > ### Comment · Reviewer_rcgL · 2025-11-26
> > **Thanks**
> >
> > Thanks for the response to my review.
> >
> > Line 117: "Credible" is used for some formal concepts in statistics, e.g. credible intervals. Maybe there is a less loaded word you can use?
> >
> > Line 470: If possible, add those alternative explanations to the paper.
> >
> > Thanks!

---

> > > ### Author Response · Authors · 2025-11-27
> > >
> > > Line 117:Thanks for pointing this out. We agree that credible in this informel sense might be misleading and have changed the wording to:
> > >
> > > > “Under Assumption 1, RDD provides inference around the threshold that is as ~~credible~~ **plausible** as  that from a randomized experiment \citep{Lee2008}.”
> > >
> > > Line 470: Thanks you for this valuable suggestion. We added this exact explaination to the text.

---

### Author Response · Authors · 2025-12-01

We would like to thank the reviewers and the area chair once more for the careful reading of our work and for the constructive feedback, which has substantially improved the paper. Below we briefly summarize how the main points raised in the reviews have been addressed in the revised version.

**1. Improved accessibility and exposition**

* Following comments by Reviewers Wgrp and MKKS about the density of the theoretical exposition, we have added two new conceptual figures (Figures 1 and 2 in the revised version) to illustrate the unit categorization (compliers, defiers, always-takers, never-takers, indecisive units) and the role of indecisive units in the MRD setting. These visualizations directly respond to the request for more intuitive explanations and make the key ideas easier to grasp for applied researchers.

**2. Additional simulations and practical roadmap**

* In response to Reviewer MKKS’s request to show more complex Boolean rules, we added an additional simulation study in Appendix B (page 18 of the revised version). This simulation considers a more complex three-rule composition and demonstrates that our subsetting approach maintains good coverage while improving precision, illustrating that the framework applies beyond the simple binary setting.
* In this new example, we also provide an explicit step-by-step description of how to subset and categorize units and how to implement the estimation in practice. This serves as a concrete “roadmap” for applied researchers, addressing Reviewer Wgrp’s request for more guidance on the practical implementation of our identification results.

**3. Clarifications for the empirical application**

* We extended the discussion of the real-world LED application, especially the interpretation of opposite-sign effects and the role of the human operator versus the automatic rule as suggested by Reviewer rcgL.

Overall, the reviews consistently acknowledge the novelty and relevance of our identification results for MRD and the value of the real-world manufacturing application. The main concerns raised by Reviewers MKKS and Wgrp were about exposition, extended simulations, and practical guidance rather than about correctness or fundamental contribution. We believe the revisions substantially address these concerns.

---

### Meta-Review · Area_Chair_dsnJ · 2026-01-12

**Summary:**

This paper presents a generalized multi-score regression discontinuity framework that extends classical RDD to multiple cutoffs, allowing flexible Boolean assignment rules and formal identification of compliers. The approach is supported by experiments on semi-synthetic data and real-world LED semiconductor manufacturing data.

**Reviewer Concerns:**

Experiments are lack of comparison and in limited settings, e.g. binary rules, and of low dimension.
The core claim is limited to its value in industrial applications, e.g. healthcare or economics.
The notation used may lack of clarity and less accessible for readers who are not familiar with the potential outcomes framework.

**Reviewer Scores:**

shall not change

---

### Decision · Program_Chairs · 2026-01-26

Reject